Largest baleen whale mass mortality during strong El Niño event is likely related to harmful toxic algal bloom

Häussermann Verena 1 2 3
http://orcid.org/0000-0002-0823-2434 Gutstein Carolina S. 4 5 6 sgcarolina@gmail.com
Bedington Michael 7
Cassis David 8
Olavarria Carlos 9 10
http://orcid.org/0000-0002-9256-7770 Dale Andrew C. 7
http://orcid.org/0000-0003-1497-364X Valenzuela-Toro Ana M. 6 11
Perez-Alvarez Maria Jose 9 12
http://orcid.org/0000-0002-0131-6627 Sepúlveda Hector H. 13
McConnell Kaitlin M. 3
Horwitz Fanny E. 14
Försterra Günter 1 3 15
1 Facultad de Ciencias Naturales, Escuela de Ciencias del Mar, Pontificia Universidad Católica de Valparaíso , Valparaíso , Chile
2 GeoBio-Center , Munich , Germany
3 Huinay Scientific Field Station , Puerto Montt, Region de los Lagos , Chile
4 Area de Patrimonio Natural, Consejo de Monumentos Nacionales , Santiago, Región Metropolitana , Chile
5 Red Paleontológica U-Chile, Laboratorio de Ontogenia y Filogenia, Departamento de Biología, Facultad de Ciencias, Universidad de Chile , Santiago, Región Metropolitana , Chile
6 Department of Paleobiology, National Museum of Natural History, Smithsonian Institution , Washington, DC , USA
7 Scottish Association for Marine Science , Oban , Scotland, UK
8 Centro de Investigación e Innovación para el Cambio Climático, Universidad Santo Tómas , Santiago , Chile
9 Centro de Investigación Eutropia , Santiago, Región Metropolitana , Chile
10 Centro de Estudios Avanzados en Zonas Aridas , La Serena , Chile
11 Ecology and Evolutionary Biology, University of California Santa Cruz , Santa Cruz, CA , USA
12 Instituto de Ecología y Biodiversidad, Facultad de Ciencias, Universidad de Chile , Santiago , Chile
13 Departamento de Geofísica, Universidad de Concepción , Concepción , Chile
14 Facultad de Ciencias Naturales y Oceanográficas, Universidad de Concepción , Concepción , Chile
15 Department of Zoology, Ludwig-Maximilians-University , Munich , Germany
Costello Mark
Electronic publication date: 2017 May 31
Publication date: 2017
Volume: 5
Electronic Location ID: e3123
Received 2016 Feb 12; Accepted 2017 Feb 26
Copyright: © 2017 Häussermann et al.
Copyright year: 2017
Copyright holder: Häussermann et al.
License: This is an open access article distributed under the terms of the Creative Commons Attribution License, which permits unrestricted use, distribution, reproduction and adaptation in any medium and for any purpose provided that it is properly attributed. For attribution, the original author(s), title, publication source (PeerJ) and either DOI or URL of the article must be cited.
License URL: https://creativecommons.org/licenses/by/4.0/

Keywords: Chilean Patagonia, Red tide, El Niño, Sei whales, Drift models, Balaenoptera borealis, Paralytic shellfish poison, Balaenopteridae, Taphonomy, Climate Change

Funding: Fondecyt Projects 1131039, 1161699 and 1150843 National Geographic Society/Waitt Grants Program #W380-15 U-REDES (Domeyko II UR-C12/1, Universidad de Chile) CONICYT Postdoctoral FONDECYT Program 3140513 CONICYT Postdoctoral FONDECYT Program 3160710 The expedition during which Verena Häussermann discovered the initial whales was funded by Fondecyt Project Nos. 1131039, 1161699 to VH and 1150843 to GF, the overflight by National Geographic Society/Waitt Grants Program #W380-15 to Carolina S. Gutstein, Verena Häussermann and Maria Jose Perez-Alvarez and the satellite image by a Pew fellowship for marine conservation to Verena Häussermann. Taphonomic analyses were funded by U-REDES (Domeyko II UR-C12/1, Universidad de Chile) to A. Vargas and Consultora Paleosuchus LTDA. Maria Jose Perez-Alvarez was funded by CONICYT Postdoctoral FONDECYT Program 3140513, Projects ICM P05-002 and PFB 023. Carolina S. Gutstein was founded by CONICYT Postdoctoral FONDECYT Program 3160710. 2016 expedition was funded by Blue Marine Foundation and Paulsen Editions Foundation to Keri Lee Pashuk (Saoirse) and Consejo de Monumentos Nacionales. The funders had no role in study design, data collection and analysis, decision to publish or preparation of the manuscript.

==============================
While large mass mortality events (MMEs) are well known for toothed whales, they have been rare in baleen whales due to their less gregarious behavior. Although in most cases the cause of mortality has not been conclusively identified, some baleen whale mortality events have been linked to bio-oceanographic conditions, such as harmful algal blooms (HABs). In Southern Chile, HABs can be triggered by the ocean–atmosphere phenomenon El Niño. The frequency of the strongest El Niño events is increasing due to climate change. In March 2015, by far the largest reported mass mortality of baleen whales took place in a gulf in Southern Chile. Here, we show that the synchronous death of at least 343, primarily sei whales can be attributed to HABs during a building El Niño. Although considered an oceanic species, the sei whales died while feeding near to shore in previously unknown large aggregations. This provides evidence of new feeding grounds for the species. The combination of older and newer remains of whales in the same area indicate that MMEs have occurred more than once in recent years. Large HABs and reports of marine mammal MMEs along the Northeast Pacific coast may indicate similar processes in both hemispheres. Increasing MMEs through HABs may become a serious concern in the conservation of endangered whale species.

Introduction

Although most populations of whales have been fully protected from industrial hunting for half a century, some were reduced to such low levels that recovery is still very slow (Baker & Clapham, 2004). Today, whales face additional threats, such as ship strikes, entanglement and by-catch, underwater noise, pollution and habitat loss (Clapham, Young & Brownell, 1999). Moreover, since ocean conditions directly influence quality and availability of the prey species of baleen whales, the effects of climate change will become a concern (Simmonds & Isaac, 2007).

Mass mortality events (MMEs) of marine mammals generally involve social species such as dolphins or sea lions, but are rare in baleen whales due to their less gregarious behavior (Perrin, Würsig & Thewissen, 2009). When MMEs have occurred in baleen whales, they have often extended over several months and large areas, involving mostly coastal whales (Table 1). In the Northeast Pacific, seven to eight times more gray whales (Eschrichtius robustus) washed ashore during the years 1999 and 2000 than is usual in such a time span. Of these, 106 died within a three-month period in Mexico (Gulland et al., 2005). In the course of 2012, 116 southern right whales (Eubalaena australis), mostly calves, washed ashore at their breeding ground in Valdés Peninsula, Argentina (Anonymous, 2015). During 2009, 46 humpback whales (Megaptera novaeangliae) stranded in Australia (Coughran, Gales & Smith, 2013) and 96 in Brazil during 2010, most of them calves and juveniles (Rowntree et al., 2013). Less frequent and much smaller in magnitude are sudden and locally restricted baleen whale mortalities. The largest of those involved 14 humpback whales, which died around Cape Cod during five weeks in Nov 1987 (Geraci et al., 1989) (Table 1). The causes of most MMEs have not been conclusively identified (Anonymous, 2015; Coughran, Gales & Smith, 2013; Gulland et al., 2005); however, paralytic shellfish poisoning (PSP) during harmful algal blooms (HABs) has been argued as one of the main likely causes (and this is also the case for other marine vertebrate mass mortalities; Geraci et al., 1989; Durbin et al., 2002; Doucette et al., 2006; Rowntree et al., 2013; Cook et al., 2015; D’Agostino et al., 2015; Wilson et al., 2015; Lefebvre et al., 2016).

Table 1 Recorded mass mortality events of baleen whales (updated from Table 1 in Rowntree et al. (2013)).

Region/site	Time span	Species	Number	Age classes	Cause of death	Source	
Caleta Buena/slight inlet, Southern Chile	Nov/Dec 1977	Rorqual	Four fresh, numerous skeletons		Unknown	M. Salas, 2015, personal communication	
Cape Cod (USA)	Five weeks (11/1987)	Humpback	14		HAB (saxitoxin)	Geraci et al. (1989)	
Upper Gulf of California (Mexico)	? (1995)	Fin, minke and bryde1	Eight		Unknown	Vidal & Gallo Reynoso (1996)	
Eastern North East Pacific	Throughout 1999	Gray	2832	Mostly adults	Malnutrition?	Gulland et al. (2005)	
Eastern North East Pacific	Throughout 2000	Gray	368	Mostly adults	Malnutrition?	Gulland et al. (2005)	
Upper Gulf of California (Mexico)	? (2009)	Unknown	10		Unknown	Rowntree et al. (2013)	
Australia	Throughout 2009	Humpback	46	Mostly calves and juveniles	Unknown	Coughran, Gales & Smith (2013)	
Brazil	Throughout 2010	Humpback	96	Mostly calves and juveniles	Unknown	Rowntree et al. (2013)	
Peninsula Valdés (Argentina)	2005–20113	Southern right	420	Mostly calves	Unknown (HAB-related? Starvation? Kelp gull harassment?)	D’Agostino et al. (2015); Wilson et al. (2015)	
Puerto Edén area (Chile)	Mar 2011	Sei and/or minke	Three		Unknown	This paper	
Estero Cono (Chile)	Mar 2012	Sei and/or minke	15		Unknown	R.M. Fischer, 2015, personal communication	
Puerto Edén area (Chile)	Jan 2014	Sei and/or minke	Five		Unknown	C. Cristie, 2015, personal communication	
Between 46 and 51°S, mainly Golfo de Penas (Chile)	Feb to early Apr 20154	Probably all sei	343	All	HAB	This paper	
Alaska/British Columbia (USA/Canada)	May/Jun 2015	Fin, humpback, gray	38		Unknown (HAB?)	NOAA (2015b)	
Notes:

1 In total, 400 cetaceans died, including eight baleen whales.

2 A total of 106 in Mexico during three months.

3 A total of 116 died during 2012.

4 A total of 271 died within one month.

Harmful algal blooms have an extended record in Southern Chile (particularly the genus Alexandrium with production of paralytic shellfish toxins (PSTs)). HABs have been of concern to fishermen and Patagonian communities since at least 1972, when the first mass intoxication was recorded (Suárez & Guzmán, 2005). Since then, the geographic region in which blooms have been detected has increased to over 1,000 km north–south extent (Molinet et al., 2003). HABs have also become more frequent, becoming annual events with blooms normally occurring in large areas during the summer and fall (Guzmán et al., 2002). Due to the danger posed by these toxins, the Chilean government funds a monitoring program with over 200 sampling stations throughout the Southern part of Chile, where phytoplankton and shellfish samples are obtained and later analyzed for the presence of microalgae and their toxins (PST, amnesic shellfish toxin (AST), diarrheic shellfish toxin (DST)) (Suárez & Guzmán, 2005). Unfortunately, mainly due to the difficulty accessing many sites, these biotoxin data are only available for a limited coastal area of Southern Chile.

Chilean Patagonia is a complex environment that hosts one of the largest and most extensive fjord regions, with a north–south extent of approximately 1,500 km (42°S–55°S), covering an area of over 240,000 km2 and with a coastline of more than 80,000 km, made up of numerous fjords, channels and islands. At the same time, this is one of the least scientifically understood marine regions of the world (Försterra, 2009; Försterra, Häussermann & Laudien, 2017). Precipitation can locally exceed 6,000 mm per year and the tidal range can exceed 7 m. The prevailing strong westerly winds make its exposed shores amongst the most wave-impacted in the world (Försterra, 2009). These factors are responsible for the inaccessibility of a large part of this region. Chilean Patagonia is subdivided into the North, Central and South Patagonian zone (for a summary of biogeography of the region see Häussermann & Försterra, 2005 and Försterra, Häussermann & Laudien, 2017). The remote area around Golfo de Penas and Taitao Peninsula (Fig. 1) is situated in the Central Patagonian Zone between 47°S and 48°S. Except for two Chilean Navy lighthouses at Cabo Raper and San Pedro, the closest human settlements are more than 200 km away (Tortel, Puerto Aysén and Puerto Edén).

Figure 1 Location of dead whales and skulls found in Chilean Patagonian.

Boat track: green (HF24), flight track: blue (HF25). (A) Golfo de Penas, (B) Golfo Tres Montes and (C) Seno Escondido.

In general, Chilean Patagonia is influenced by the West Wind Drift, a large-scale eastward (onshore) flow which diverges at the coast to form the northward Humboldt Current and the southward Cape Horn Current (Thiel et al., 2007). The fjordic nature of the coastline produces significant local complexity, with many inlets and dispersed freshwater sources. High productivity in these coastal waters (Fig. 2) is driven by the availability of both terrestrial nutrients, carried by large rivers originating at the Northern and Southern Patagonian Ice Fields, and marine nutrients (González et al., 2010; Torres et al., 2014). While this region experiences coastal winds that favor net coastal downwelling, intermittent and/or localized upwelling, in particular in summer and North of Taitao Peninsula (47°S), is expected to enhance the supply of marine nutrients to coastal waters, and the relative balance between upwelling and downwelling varies from year to year.

Figure 2 Satellite image (MODIS Aqua) showing the concentration of chlorophyll a on Mar 23, 2015.

Areas where most whales were found are circled.

During a vessel-based scuba diving expedition, “Huinay Fiordos 24” (HF24), focused on benthic fauna between Golfo Tres Montes (Northern Golfo de Penas, 46°30′W) and Puerto Eden (49°S), dead baleen whales and skeletal remains were discovered south of Golfo de Penas and at Golfo Tres Montes. Here, we describe by far the largest ever-recorded MME of baleen whales at one time and place. Our analyses focus on the location and cause of the mortality.

Materials and Methods

Field surveys

The vessel-based HF24 scuba diving expedition, from Apr 15 to May 8, 2015, aimed to inventory the benthic fauna of the area between Golfo Tres Montes (Northern Golfo de Penas, 46°30′W) and Puerto Edén (49°S). By chance, VH and her team discovered recently dead baleen whales and skeletal remains in and close to the entrance of the 14 km long Estero Slight and in the Canal Castillo situated 235 km to the south (Figs. 1 and 3; Table 2). Georeferences and photographs of different views were taken, whales measured, and species and sex identified whenever possible. Between May 25 and 31, the Chilean Fisheries Service (SERNAPESCA), with the support of the Chilean Navy (Armada) and the Criminal Investigation Department of the Civil Police (PDI), organized a vessel-based trip to the location of the dead whales in Estero Slight to investigate possible anthropogenic reasons behind the mortality. During this trip, genetic samples for species identification were taken, one ear bone was extracted and stomach and intestine contents of two whales were tested for presence of PST and AST (Fiscalía de Aysén, 2015). During a subsequent aerial survey, on-board a high wing airplane Cessna 206, between Jun 23 and 27, 2015, three of us (CG, VH and FH) surveyed the coasts along the shores of Golfo de Penas. This aerial survey covered the coastal area between the Jungfrauen Islands (48°S) and Seno Newman (46°39′S) from altitudes between 100 and 850 m and at speeds between 100 and 200 km/h (Figs. 1 and 4; Table 2). Due to limited flying time (unstable weather conditions and the inability to refuel in the area), data collection was focused on counting whale carcasses, recording GPS positions and taking photographs. A GoPro camera filmed continuously until reaching Seno Newman. The researchers on the flight counted carcasses and marked their coordinates while an audio recorder captured the carcass number, position, orientation, photo number, photographer and geomorphology of the beach. Whale counts were repeated in all areas except Seno Newman due to adverse weather conditions. Since there are no landing opportunities in this remote and unpopulated area, it was not possible to take samples or close-up photos, or to search for additional whale bones.

Figure 3 Documented whale carcasses and skeletal remains during a vessel survey in Apr 21, 2015 in Caleta Buena, Estero Slight.

(A and B) Skeletal remains. (C) Recently dead sei whale. Photos: Keri-Lee Pashuk, all rights reserved.

Table 2 List of whale carcasses, their degree of decomposition/disarticulation, location and date of finding.

Date	Locality	Whale ID	Latitude	Longitude	State of decomposition	Time at sea	Carcass position	Beach type	Species	Sex	
HF24 expedition	
Apr 21, 2015	West of Isla Centro	1	46°43.158′S	75°22.09′W	1		Lateral-up	Rocky	Balaenoptera borealis		
Apr 21, 2015	West of Isla Centro	2	46°43.069′S	75°22.553′W	1		Lateral-up	Rocky	Balaenoptera borealis	Male	
Apr 21, 2015	West of Isla Centro	3	46°43.095′S	75°22.759′W	1		Lateral-up	Rocky	Balaenoptera borealis		
Apr 21, 2015	West of Isla Centro	4	46°43.561′S	75°26.079′W	1			Rocky	Balaenopteridae		
Apr 21, 2015	Caleta Buena	5	46°46.92′S	75°30.057′W	1		Ventral-up	Floating	Balaenoptera borealis	Female	
Apr 21, 2015	Caleta Buena	6	46°47.25′S	75°29.872′W	1		Lateral-up	Floating	Balaenoptera borealis	Male	
Apr 21, 2015	Caleta Buena	7	46°47.248′S	75°29.876′W	1		Ventral-up	Floating			
Apr 21, 2015	Caleta Buena	8	46°47.275′S	75°29.837′W	1		Lateral-up	Rocky	Balaenoptera borealis	Male	
Apr 21, 2015	Caleta Buena	9	46°47.268′S	75°29.82′W	1		Lateral-up	Rocky	Balaenoptera borealis		
Apr 21, 2015	Caleta Buena	10	46°47.264′S	75°29.807′W	1		Lateral-up	Rocky	Balaenoptera borealis	Female	
Apr 21, 2015	Caleta Buena	11	46°47.261′S	75°29.798′W	1		Lateral-up	Rocky	Balaenoptera borealis	Female	
Apr 21, 2015	Caleta Buena	12	46°47.253′S	75°29.789′W	1		Lateral-up	Rocky	Balaenoptera borealis	Male	
Apr 21, 2015	Caleta Buena	13	46°47.249′S	75°29.787′W	3		Ventral-up	Rocky			
Apr 21, 2015	Caleta Buena	14	46°47.258′S	75°29.8′W	3		Ventral-up	Floating			
Apr 21, 2015	Caleta Buena	15	46°47.261′S	75°29.812′W	3		Ventral-up	Floating			
Apr 22, 2015	Estero Slight	16	46°47.135′S	75°32.269′W	2		Lateral-up	Rocky	Balaenoptera borealis		
Apr 22, 2015	Estero Slight	17	46°47.214′S	75°34.332′W	3		Ventral-up	Rocky			
Apr 22, 2015	Estero Slight	18	46°47.817′S	75°32.788′W	1		Lateral-up	Rocky	Balaenoptera borealis	Female	
Apr 22, 2015	Estero Slight	19	46°47.951′S	75°32.973′W	2			Floating	Balaenoptera borealis		
Apr 22, 2015	Estero Slight	20	46°48.023′S	75°33.055′W	2			Rocky	Balaenoptera borealis		
Apr 22, 2015	Estero Slight	21	46°48.264′S	75°33.425′W	2		Ventral-up	Rocky	Balaenoptera borealis	Female	
Apr 22, 2015	Estero Slight	22	46°48.51′S	75°33.909′W	1		Lateral-up	Rocky	Balaenoptera borealis	Female	
Apr 22, 2015	Estero Slight	23	46°48.508′S	75°33.914′W	1		Lateral-up	Rocky	Balaenoptera borealis	Male	
Apr 22, 2015	Estero Slight	24	46°48.515′S	75°34.668′W	1		Lateral-up	Sandy	Balaenopteridae		
Apr 22, 2015	Estero Slight	25	46°48.511′S	75°34.684′W	3		Dorsal up	Sandy	Balaenoptera borealis	Female	
Apr 22, 2015	Estero Slight	26	46°48.206′S	75°34.905′W	3		Ventral-up	Rocky			
Apr 22, 2015	Estero Slight	27	46°48.204′S	75°34.909′W	1		Lateral-up	Rocky	Balaenoptera borealis		
Apr 22, 2015	Estero Slight	28	46°48.09′S	75°34.9′W	3		Ventral-up	Rocky			
Apr 22, 2015	Estero Slight	29	46°48.01′S	75°34.909′W	3		Lateral-up	Rocky			
Apr 22, 2015	Estero Slight	30	46°48.008′S	75°34.902′W	3		Lateral-up	Rocky			
Apr 22, 2015	Estero Slight	31	46°47.919′S	75°34.86′W	1		Lateral-up	Floating	Balaenoptera borealis	Female	
Apr 22, 2015	Estero Slight	32	46°47.642′S	75°34.753′W	1		Lateral-up	Rocky	Balaenoptera borealis		
Apr 22, 2015	Estero Slight	33	46°47.538′S	75°34.651′W	1		Lateral-up	Rocky	Balaenopteridae		
Apr 22, 2015	Estero Slight	34	46°47.442′S	75°34.463′W	3		Lateral-up	Rocky			
Apr 22, 2015	Estero Slight	35	46°46.173′S	75°33.247′W	1		Ventral-up	Rocky	Balaenoptera borealis	Male	
Apr 22, 2015	Estero Slight	36	48°46.002′S	75°33.066′W	3		Ventral-up	Rocky			
Apr 22, 2015	Estero Slight/Baja Julio	37	48°45.626′S	75°31.102′W	2			Floating			
Apr 22, 2015	Estero Slight/Baja Julio	38	48°45.530′S	75°30.962′W	1		Lateral-up	Rocky	Balaenopteridae		
Apr 22, 2015	Estero Slight/Baja Julio	39	46°45.205′S	75°30.75′W	1		Lateral-up	Rocky	Balaenoptera borealis		
Apr 22, 2015	Estero Slight/Baja Julio	40	46°45.008′S	75°30.674′W	1		Lateral-up	Rocky	Balaenoptera borealis		
Apr 22, 2015	Islote Amarillo	41	46°40.967′S	75°27.983′W	1		Lateral-up	Rocky	Balaenopteridae		
Apr 22, 2015	Islote Amarillo	42	46°40.722′S	75°27.21′W	1		Lateral-up	Rocky	Balaenopteridae		
Apr 22, 2015	Isla Esmeralda	43	48°48.08′S	75°24.29′W							
Apr 22, 2015	Isla Hyatt	44	48°47.95′S	75°26.45′W							
Apr 22, 2015	Isla Hyatt	45	48°47.3′S	75°26.13′W							
Apr 22, 2015	Isla Hyatt	46	48°47.26′S	75°26.01′W							
Apr 22, 2015	Isla Hyatt	47	48°47.19′S	75°25.91′W							
HF25 expedition	
Jun 23, 2015	Jungfrauen group	48	47°32.29′S	74°32.484′W	2	2	Lateral-up	Rocky			
Jun 23, 2015	Jungfrauen group	49	47°36.16′S	74°34.997′W				Floating			
Jun 24, 2015	Jungfrauen group	50	48°3.874′S	75°1.788′W				Floating	Balaenopteridae		
Jun 24, 2015	Jungfrauen group	51	48°3.875′S	75°1.791′W				Floating	Balaenopteridae		
Jun 24, 2015	Jungfrauen group	52	48°4.209′S	75°1.052′W				Rocky			
Jun 24, 2015	Jungfrauen group	53	48°3.361′S	75°7.514′W				Rocky			
Jun 24, 2015	Jungfrauen group	54	47°59.048′S	75°15.302′W	2	2	Lateral-up	Rocky			
Jun 24, 2015	Jungfrauen group	55	47°57.402′S	75°15.671′W	2	2		Rocky			
Jun 24, 2015	Jungfrauen group	56	47°57.554′S	75°14.56′W	2	1		Rocky			
Jun 24, 2015	Jungfrauen group	57	47°56.28′S	75°14.706′W	2	1		Rocky			
Jun 24, 2015	Jungfrauen group	58	47°51.025′S	75°13.345′W	1	1		Rocky	Balaenopteridae		
Jun 24, 2015	Jungfrauen group	59	47°50.923′S	75°12.912′W	1	1		Rocky			
Jun 24, 2015	Jungfrauen group	60	47°50.701′S	75°12.218′W	2	1	Lateral-up	Rocky			
Jun 24, 2015	Jungfrauen group	61	47°50.799′S	75°13.279′W	2	1	Ventral-up	Rocky			
Jun 24, 2015	Jungfrauen group	62	47°48.885′S	75°12.317′W	2			Sandy			
Jun 24, 2015	Jungfrauen group	63	47°48.598′S	75°12.183′W	1	1	Lateral-up	Sandy	Balaenopteridae		
Jun 24, 2015	Jungfrauen group	64	47°52.994′S	75°11.915′W				Rocky			
Jun 24, 2015	Jungfrauen group	65	47°52.766′S	75°11.704′W				Rocky			
Jun 24, 2015	Jungfrauen group	66	47°53.019′S	75°9.343′W	2			Rocky			
Jun 24, 2015	Jungfrauen group	67	47°53.004′S	75°9.316′W	2	1		Rocky			
Jun 24, 2015	Jungfrauen group	68	47°52.409′S	75°8.578′W				Sandy	Balaenopteridae		
Jun 24, 2015	Jungfrauen group	69	47°51.775′S	75°4.472′W	1	1	Lateral-up	Rocky	Balaenopteridae		
Jun 24, 2015	Jungfrauen group	70	47°51.527′S	75°3.374′W				Rocky	Balaenopteridae		
Jun 24, 2015	Jungfrauen group	71	47°49.123′S	75°3.696′W	2	1	Ventral-up	Rocky	Balaenopteridae		
Jun 24, 2015	Jungfrauen group	72	47°49.698′S	75°59.956′W				Rocky	Balaenopteridae		
Jun 24, 2015	Jungfrauen group	73	47°47.558′S	74°58.11′W			Lateral-up	Rocky	Balaenopteridae		
Jun 24, 2015	Jungfrauen group	74	47°47.474′S	74°58.107′W	2	1	Lateral-up	Rocky	Balaenopteridae		
Jun 24, 2015	Jungfrauen group	75	47°47.089′S	74°58.445′W				Rocky			
Jun 24, 2015	Jungfrauen group	76	47°17.614′S	74°22.75′W				Sandy			
Jun 24, 2015	Jungfrauen group	77	47°17.454′S	74°22.494′W				Sandy			
Jun 24, 2015	San Quintin bay I	78	46°50.51′S	74°37.41′W				Sandy			
Jun 24, 2015	San Quintin bay I	79	46°49.324′S	74°35.952′W				Sandy			
Jun 24, 2015	San Quintin bay I	80	46°49.905′S	74°36.359′W			Lateral-up	Sandy	Balaenoptera borealis		
Jun 24, 2015	San Quintin bay I	81	46°49.973′S	74°36.381′W			Lateral-up	Sandy	Balaenoptera borealis		
Jun 24, 2015	San Quintin bay I	82	46°50.495′S	74°36.442′W	1	1	Lateral-up	Rocky	Balaenopteridae		
Jun 24, 2015	San Quintin bay I	83	46°50.488′S	74°36.428′W	1	1	Lateral-up	Rocky	Balaenopteridae		
Jun 24, 2015	San Quintin bay I	84	46°50.476′S	74°36.304′W			Ventral-up	Sandy–rocky			
Jun 24, 2015	San Quintin bay I	85	46°50.474′S	74°36.288′W	1	1	Lateral-up	Sandy–rocky	Balaenopteridae		
Jun 24, 2015	San Quintin bay I	86	46°50.476′S	74°36.262′W	1	1	Ventral-up	Sandy–rocky			
Jun 24, 2015	San Quintin bay I	87	46°50.47′S	74°36.128′W	2		Lateral-up	rocky	Balaenopteridae		
Jun 24, 2015	San Quintin bay I	88	46°50.467′S	74°36.104′W	1	1	Lateral-up	rocky			
Jun 24, 2015	San Quintin bay I	89	46°50.444′S	74°36.02′W	2	1	Lateral-up	Sandy–rocky			
Jun 24, 2015	San Quintin bay I	90	46°50.437′S	74°35.943′W	1	1	Lateral-up	Sandy	Balaenopteridae		
Jun 24, 2015	San Quintin bay I	91	46°50.431′S	74°35.931′W	1	1	Lateral-up	Sandy	Balaenopteridae		
Jun 24, 2015	San Quintin bay I	92	46°50.428′S	74°35.925′W	1	1	Lateral-up	Sandy	Balaenopteridae		
Jun 24, 2015	San Quintin bay I	93	46°50.422′S	74°35.926′W	1	1	Lateral-up	Sandy	Balaenopteridae		
Jun 24, 2015	San Quintin bay I	94	46°50.405′S	74°35.924′W	1	1	Lateral-up	Sandy	Balaenopteridae		
Jun 24, 2015	San Quintin bay I	95	46°50.404′S	74°35.921′W	1	1	Lateral-up	Sandy			
Jun 24, 2015	San Quintin bay I	96	46°50.371′S	74°35.951′W	2	1	Lateral-up	Sandy			
Jun 24, 2015	San Quintin bay I	97	46°50.357′S	74°35.96′W	1	1	Lateral-up	Sandy	Balaenopteridae		
Jun 24, 2015	San Quintin bay I	98	46°50.355′S	74°35.957′W	1	1	Lateral-up	Sandy	Balaenopteridae		
Jun 24, 2015	San Quintin bay I	99	46°50.353′S	74°35.96′W	1	1	Lateral-up	Sandy	Balaenopteridae		
Jun 24, 2015	San Quintin bay I	100	46°50.326′S	74°36.22′W	1	1	Lateral-up	Rocky	Balaenopteridae		
Jun 24, 2015	San Quintin bay I	101	46°50.322′S	74°36.024′W	1	1	Ventral-up	Rocky	Balaenopteridae		
Jun 24, 2015	San Quintin bay I	102	46°50.285′S	74°36.188′W	1	1	Lateral-up	Sandy	Balaenopteridae		
Jun 24, 2015	San Quintin bay I	103	46°50.256′S	74°36.102′W	1	1		Rocky			
Jun 24, 2015	San Quintin bay I	104	46°50.254′S	74°36.094′W	2	1		Rocky			
Jun 24, 2015	San Quintin bay I	105	46°50.23′S	74°36.073′W	1	1	Lateral-up	Sandy–rocky	Balaenopteridae		
Jun 24, 2015	San Quintin bay I	106	46°50.1′S	74°36.194′W	1	1	Lateral-up	Rocky	Balaenopteridae		
Jun 24, 2015	San Quintin bay I	107	46°50.243′S	74°35.836′W	2	1	Ventral-up	Sandy			
Jun 24, 2015	San Quintin bay I	108	46°50.247′S	74°35.834′W	2	1	Ventral-up	Sandy			
Jun 24, 2015	San Quintin bay I	109	46°50.251′S	74°35.652′W	3	1	Ventral-up	Sandy			
Jun 24, 2015	San Quintin bay I	110	46°50.258′S	74°35.639′W	2	1	Ventral-up	Sandy			
Jun 24, 2015	San Quintin bay I	111	46°50.212′S	74°35.585′W	1	1	Ventral-up	Sandy–rocky	Balaenopteridae		
Jun 24, 2015	San Quintin bay I	112	46°50.229′S	74°35.513′W	2	1	Lateral-up	Sandy–rocky			
Jun 24, 2015	San Quintin bay I	113	46°50.222′S	74°35.483′W	2			Rocky			
Jun 24, 2015	San Quintin bay I	114	46°50.214′S	74°35.429′W	1	1	Ventral-up	Rocky	Balaenopteridae		
Jun 24, 2015	San Quintin bay I	115	46°50.195′S	74°35.315′W	1	1	Ventral-up	Rocky			
Jun 24, 2015	San Quintin bay I	116	46°50.184′S	74°35.18′W			Ventral-up	Sandy–rocky	Balaenopteridae		
Jun 24, 2015	San Quintin bay I	117	46°50.172′S	74°35.1′W	2	1	Ventral-up	Rocky			
Jun 24, 2015	San Quintin bay I	118	46°50.126′S	74°34.995′W	2	1		Sandy–rocky			
Jun 24, 2015	San Quintin bay I	119	46°50.122′S	74°34.894′W				Sandy–rocky			
Jun 24, 2015	San Quintin bay I	120	46°49.958′S	74°34.433′W				Rocky			
Jun 24, 2015	San Quintin bay I	121	46°49.928′S	74°34.459′W	2	1		Rocky			
Jun 24, 2015	San Quintin bay I	122	46°49.902′S	74°34.385′W	2			Rocky			
Jun 24, 2015	San Quintin bay I	123	46°49.879′S	74°34.158′W			Ventral-up	Sandy–rocky			
Jun 24, 2015	San Quintin bay I	124	46°50.482′S	74°38.058′W	1	1	Lateral-up	Rocky	Balaenopteridae		
Jun 24, 2015	San Quintin bay II	125	46°48.956′S	74°39.394′W	1	1	Lateral-up	Sandy	Balaenopteridae		
Jun 24, 2015	San Quintin bay II	126	46°49.207′S	74°39.756′W			Lateral-up	Sandy	Balaenopteridae		
Jun 24, 2015	San Quintin bay II	127	46°49.145′S	74°40.03′W			Lateral-up	Sandy	Balaenopteridae		
Jun 24, 2015	San Quintin bay II	128	46°49.299′S	74°40.244′W			Lateral-up	Rocky	Balaenopteridae		
Jun 24, 2015	San Quintin bay II	129	46°49.136′S	74°40.346′W			Lateral-up	Sandy	Balaenopteridae		
Jun 24, 2015	San Quintin bay II	130	46°49.134′S	74°40.346′W			Ventral-up	Sandy	Balaenopteridae		
Jun 24, 2015	San Quintin bay II	131	46°49.117′S	74°40.317′W			Lateral-up	Sandy	Balaenopteridae		
Jun 24, 2015	San Quintin bay II	132	46°49.12′S	74°40.324′W			Lateral-up	Sandy	Balaenopteridae		
Jun 24, 2015	San Quintin bay II	133	46°48.872′S	74°40.634′W				Rocky	Balaenopteridae		
Jun 24, 2015	San Quintin bay II	134	46°49.026′S	74°40.594′W	1	1	Lateral-up	Rocky	Balaenopteridae		
Jun 24, 2015	San Quintin bay II	135	46°49.017′S	74°40.617′W	1	1		Rocky	Balaenopteridae		
Jun 24, 2015	San Quintin bay II	136	46°49.111′S	74°40.713′W	1	1	Ventral-up	Rocky	Balaenopteridae		
Jun 24, 2015	San Quintin bay II	137	46°49.109′S	74°40.727′W	1	1	Lateral-up	Rocky	Balaenopteridae		
Jun 24, 2015	San Quintin bay II	138	46°49.243′S	74°40.792′W	1	1		Rocky	Balaenopteridae		
Jun 24, 2015	San Quintin bay II	139	46°49.218′S	74°40.821′W				Rocky	Balaenopteridae		
Jun 24, 2015	San Quintin bay II	140	46°49.182′S	74°40.863′W	1			Rocky	Balaenopteridae		
Jun 24, 2015	San Quintin bay II	141	46°49.185′S	74°40.893′W	1	1		Rocky	Balaenopteridae		
Jun 24, 2015	San Quintin bay II	142	46°49.155′S	74°41.014′W			Lateral-up	Rocky	Balaenopteridae		
Jun 24, 2015	San Quintin bay II	143	46°49.146′S	74°41.118′W			Lateral-up	Rocky	Balaenopteridae		
Jun 24, 2015	San Quintin bay II	144	46°48.985′S	74°41.307′W			Lateral-up	Rocky	Balaenopteridae		
Jun 24, 2015	San Quintin bay II	145	46°49.003′S	74°41.312′W			Lateral-up	Rocky	Balaenopteridae		
Jun 24, 2015	San Quintin bay II	146	46°49.008′S	74°41.313′W			Lateral-up	Rocky	Balaenopteridae		
Jun 24, 2015	San Quintin bay II	147	46°49.028′S	74°41.327′W			Lateral-up	Rocky	Balaenopteridae		
Jun 24, 2015	San Quintin bay II	148	46°49.061′S	74°41.359′W				Rocky			
Jun 24, 2015	San Quintin bay II	149	46°49.104′S	74°41.404′W				Rocky	Balaenopteridae		
Jun 24, 2015	San Quintin bay II	150	46°49.027′S	74°41.441′W				Rocky	Balaenopteridae		
Jun 24, 2015	San Quintin bay II	151	46°48.909′S	74°41.55′W				Floating			
Jun 24, 2015	San Quintin bay II	152	46°48.87′S	74°41.539′W				Floating	Balaenopteridae		
Jun 24, 2015	San Quintin bay II	153	46°48.645′S	74°41.697′W				Sandy			
Jun 24, 2015	San Quintin bay II	154	46°48.691′S	74°41.584′W				Sandy			
Jun 24, 2015	San Quintin bay II	155	46°46.879′S	74°46.086′W			Lateral-up	Sandy	Balaenopteridae		
Jun 24, 2015	San Quintin bay II	156	46°49.78′S	74°32.109′W				Rocky			
Jun 24, 2015	Seno Newman	157	46°43.813′S	74°57.964′W	2	2		Sandy			
Jun 24, 2015	Seno Newman	158	46°41.327′S	75°0.753′W	1	1	Lateral-up	Sandy	Balaenopteridae		
Jun 24, 2015	Seno Newman	159	46°37.458′S	75°2.434′W	2			Sandy–rocky			
Jun 24, 2015	Seno Newman	160	46°37.415′S	75°2.637′W	1	1		Sandy–rocky	Balaenopteridae		
Jun 24, 2015	Seno Newman	161	46°37.415′S	75°2.635′W	1	1		Sandy–rocky			
Jun 24, 2015	Seno Newman	162	46°36.941′S	75°2.113′W	2	1		Sandy			
Jun 24, 2015	Seno Newman	163	46°36.918′S	75°2.082′W	1	1	Lateral-up	Sandy	Balaenopteridae		
Jun 24, 2015	Seno Newman	164	46°36.854′S	75°2.004′W	1	1	Lateral-up	Sandy	Balaenopteridae		
Jun 24, 2015	Seno Newman	165	46°36.756′S	75°1.976′W				Rocky			
Jun 24, 2015	Seno Newman	166	46°36.539′S	75°1.71′W	1	1	Lateral-up	Sandy–rocky	Balaenopteridae		
Jun 24, 2015	Seno Newman	167	46°36.441′S	75°2.075′W	1	1	Ventral-up	Sandy			
Jun 24, 2015	Seno Newman	168	46°36.369′S	75°1.672′W				Floating			
Jun 24, 2015	Seno Newman	169	46°35.82′S	75°1.375′W	1	1	Ventral-up	Rocky	Balaenopteridae		
Jun 24, 2015	Seno Newman	170	46°35.377′S	75°1.041′W	1	1	Ventral-up	Rocky			
Jun 24, 2015	Seno Newman	171	46°35.161′S	75°0.66′W				Sandy	Balaenopteridae		
Jun 24, 2015	Seno Newman	172	46°35.087′S	75°0.513′W	1			Sandy–rocky	Balaenopteridae		
Jun 24, 2015	Seno Newman	173	46°35.089′S	75°0.49′W	1			Sandy			
Jun 24, 2015	Seno Newman	174	46°35.083′S	75°0.42′W	1	1		Floating	Balaenopteridae		
Jun 24, 2015	Seno Newman	175	46°35.085′S	74°59.71′W			Lateral-up	Rocky	Balaenopteridae		
Jun 24, 2015	Seno Newman	176	46°34.88′S	74°59.475′W				Sandy	Balaenopteridae		
Jun 24, 2015	Seno Newman	177	46°34.794′S	74°59.426′W				Sandy	Balaenopteridae		
Jun 24, 2015	Seno Newman	178	46°34.449′S	74°59.313′W				Sandy–rocky	Balaenopteridae		
Jun 24, 2015	Seno Newman	179	46°33.721′S	74°59.271′W	2		Ventral-up	Sandy			
Jun 24, 2015	Seno Newman	180	46°33.501′S	74°59.192′W	2	1		Sandy–rocky			
Jun 24, 2015	Seno Newman	181	46°33.125′S	74°58.681′W	2			Rocky			
Jun 24, 2015	Seno Newman	182	46°33.12′S	74°58.674′W	1	1		Rocky			
Jun 24, 2015	Seno Newman	183	46°32.939′S	74°58.52′W	1	1		Sandy	Balaenopteridae		
Jun 24, 2015	Seno Newman	184	46°32.521′S	74°57.707′W	2			Rocky			
Jun 24, 2015	Seno Newman	185	46°32.473′S	74°57.635′W	2	1		Rocky			
Jun 24, 2015	Seno Newman	186	46°32.424′S	74°57.582′W	2	1		Rocky			
Jun 24, 2015	Seno Newman	187	46°32.388′S	74°57.532′W	2		Lateral-up	Rocky	Balaenopteridae		
Jun 24, 2015	Seno Newman	188	46°32.346′S	74°57.469′W	1	1	Lateral-up	Sandy–rocky	Balaenopteridae		
Jun 24, 2015	Seno Newman	189	46°32.348′S	74°57.469′W	1	1	Lateral-up	Sandy–rocky	Balaenopteridae		
Jun 24, 2015	Seno Newman	190	46°32.267′S	74°57.188′W	2	1	Ventral-up	Sandy–rocky	Balaenopteridae		
Jun 24, 2015	Seno Newman	191	46°32.096′S	74°57.303′W	1	1		Sandy			
Jun 24, 2015	Seno Newman	192	46°32.07′S	74°57.254′W	1	1	Lateral-up	Sandy–rocky	Balaenopteridae		
Jun 24, 2015	Seno Newman	193	46°32.068′S	74°57.247′W	1	1	Lateral-up	Sandy–rocky	Balaenopteridae		
Jun 24, 2015	Seno Newman	194	46°32.027′S	74°57.153′W	1	1	Lateral-up	Sandy–rocky	Balaenopteridae		
Jun 24, 2015	Seno Newman	195	46°31.998′S	74°57.106′W	1	1	Lateral-up	Sandy–rocky	Balaenopteridae		
Jun 24, 2015	Seno Newman	196	46°31.919′S	74°57.006′W	1	1	Lateral-up	Rocky	Balaenopteridae		
Jun 24, 2015	Seno Newman	197	46°31.852′S	74°56.936′W	1	1	Lateral-up	Rocky	Balaenopteridae		
Jun 24, 2015	Seno Newman	198	46°31.829′S	74°56.922′W	1	1	Lateral-up	Rocky	Balaenopteridae		
Jun 24, 2015	Seno Newman	199	46°31.721′S	74°56.839′W	1	1	Lateral-up	Sandy–rocky	Balaenopteridae		
Jun 24, 2015	Seno Newman	200	46°31.592′S	74°56.733′W	1	1	Lateral-up	Rocky	Balaenopteridae		
Jun 24, 2015	Seno Newman	201	46°31.461′S	74°56.568′W	2	1	Lateral-up	Sandy			
Jun 24, 2015	Seno Newman	202	46°31.311′S	74°56.537′W	1	1	Lateral-up	Sandy–rocky	Balaenopteridae		
Jun 24, 2015	Seno Newman	203	46°31.304′S	74°56.525′W	2	1	Lateral-up	Sandy–rocky	Balaenopteridae		
Jun 24, 2015	Seno Newman	204	46°31.265′S	74°56.489′W	2	1	Lateral-up	Rocky	Balaenopteridae		
Jun 24, 2015	Seno Newman	205	46°31.055′S	74°56.197′W	1	1	Lateral-up	Sandy–rocky	Balaenopteridae		
Jun 24, 2015	Seno Newman	206	46°30.974′S	74°56.093′W	2	1	Ventral-up	Sandy–rocky	Balaenopteridae		
Jun 24, 2015	Seno Newman	207	46°30.948′S	74°56.065′W	2	1	Lateral-up	Sandy–rocky			
Jun 24, 2015	Seno Newman	208	46°30.866′S	74°55.959′W	1	1	Lateral-up	Sandy–rocky	Balaenopteridae		
Jun 24, 2015	Seno Newman	209	46°30.859′S	74°55.953′W	2	1	Lateral-up	Sandy–rocky	Balaenopteridae		
Jun 24, 2015	Seno Newman	210	46°30.824′S	74°55.907′W	2	1	Lateral-up	Rocky	Balaenopteridae		
Jun 24, 2015	Seno Newman	211	46°30.757′S	74°55.817′W	1	1	Ventral-up	Rocky	Balaenopteridae		
Jun 24, 2015	Seno Newman	212	46°30.702′S	74°55.734′W	1	1	Ventral-up	Rocky	Balaenopteridae		
Jun 24, 2015	Seno Newman	213	46°30.709′S	74°55.689′W	1	1	Lateral-up	Sandy	Balaenopteridae		
Jun 24, 2015	Seno Newman	214	46°30.707′S	74°55.674′W	1	1	Lateral-up	Sandy	Balaenopteridae		
Jun 24, 2015	Seno Newman	215	46°30.662′S	74°55.593′W	3		Ventral-up	Rocky	Balaenopteridae		
Jun 24, 2015	Seno Newman	216	46°30.624′S	74°55.439′W	1	1	Lateral-up	Sandy–rocky	Balaenopteridae		
Jun 24, 2015	Seno Newman	217	46°30.627′S	74°55.432′W	2	1	Lateral-up	Sandy–rocky	Balaenopteridae		
Jun 24, 2015	Seno Newman	218	46°30.629′S	74°55.425′W	2	1	Lateral-up	Sandy–rocky	Balaenopteridae		
Jun 24, 2015	Seno Newman	219	46°30.632′S	74°55.419′W	2		Lateral-up	Sandy–rocky	Balaenopteridae		
Jun 24, 2015	Seno Newman	220	46°30.63′S	74°55.411′W	1	1	Lateral-up	Sandy–rocky	Balaenopteridae		
Jun 24, 2015	Seno Newman	221	46°30.627′S	74°55.368′W			Lateral-up	Sandy–rocky	Balaenopteridae		
Jun 24, 2015	Seno Newman	222	46°30.618′S	74°55.338′W			Lateral-up	Sandy–rocky	Balaenopteridae		
Jun 24, 2015	Seno Newman	223	46°30.191′S	74°55.327′W			Lateral-up	Rocky	Balaenopteridae		
Jun 24, 2015	Seno Newman	224	46°30.093′S	74°55.297′W	1	1	Lateral-up	Sandy	Balaenopteridae		
Jun 24, 2015	Seno Newman	225	46°30.054′S	74°55.243′W	1	1	Lateral-up	Rocky	Balaenopteridae		
Jun 24, 2015	Seno Newman	226	46°29.992′S	74°55.167′W	1	1	Lateral-up	Rocky	Balaenopteridae		
Jun 24, 2015	Seno Newman	227	46°29.984′S	74°55.165′W	1	1	Lateral-up	Rocky	Balaenopteridae		
Jun 24, 2015	Seno Newman	228	46°29.975′S	74°55.164′W	1	1	Lateral-up	Rocky	Balaenopteridae		
Jun 24, 2015	Seno Newman	229	46°29.925′S	74°55.167′W	1	1	Lateral-up	Rocky	Balaenopteridae		
Jun 24, 2015	Seno Newman	230	46°29.895′S	74°55.166′W	2		Lateral-up	Rocky	Balaenopteridae		
Jun 24, 2015	Seno Newman	231	46°29.742′S	74°55.164′W	1	1	Lateral-up	Rocky	Balaenopteridae		
Jun 24, 2015	Seno Newman	232	46°29.329′S	74°55.094′W	1	1	Lateral-up	Floating			
Jun 24, 2015	Seno Newman	233	46°29.385′S	74°54.993′W			Lateral-up	Sandy	Balaenopteridae		
Jun 24, 2015	Seno Newman	234	46°29.32′S	74°54.924′W			Lateral-up	Rocky	Balaenopteridae		
Jun 24, 2015	Seno Newman	235	46°29.218′S	74°54.888′W	1	1	Ventral-up	Sandy–rocky	Balaenopteridae		
Jun 24, 2015	Seno Newman	236	46°29.137′S	74°54.821′W	1	1	Lateral-up	Sandy–rocky	Balaenopteridae		
Jun 24, 2015	Seno Newman	237	46°29.131′S	74°54.818′W	2	1	Lateral-up	Sandy–rocky	Balaenopteridae		
Jun 24, 2015	Seno Newman	238	46°29.124′S	74°54.813′W	1	1	Lateral-up	Sandy–rocky	Balaenopteridae		
Jun 24, 2015	Seno Newman	239	46°29.106′S	74°54.809′W	1	1	Lateral-up	Sandy–rocky	Balaenopteridae		
Jun 24, 2015	Seno Newman	240	46°29.086′S	74°54.803′W	1	1	Lateral-up	Sandy	Balaenopteridae		
Jun 24, 2015	Seno Newman	241	46°29.066′S	74°54.813′W	1	1	Lateral-up	Sandy	Balaenopteridae		
Jun 24, 2015	Seno Newman	242	46°28.991′S	74°54.825′W	1	1	Lateral-up	Rocky	Balaenopteridae		
Jun 24, 2015	Seno Newman	243	46°28.911′S	74°54.822′W	1	1	Lateral-up	Rocky	Balaenopteridae		
Jun 24, 2015	Seno Newman	244	46°28.887′S	74°54.826′W			Lateral-up	Rocky	Balaenopteridae		
Jun 24, 2015	Seno Newman	245	46°28.812′S	74°54.831′W			Lateral-up	Rocky	Balaenopteridae		
Jun 24, 2015	Seno Newman	246	46°28.761′S	74°54.83′W			Lateral-up	Rocky	Balaenopteridae		
Jun 24, 2015	Seno Newman	247	46°28.705′S	74°54.828′W	1		Lateral-up	Rocky	Balaenopteridae		
Jun 24, 2015	Seno Newman	248	46°28.658′S	74°54.828′W	1	1	Lateral-up	Rocky	Balaenopteridae		
Jun 24, 2015	Seno Newman	249	46°28.654′S	74°54.831′W	1		Lateral-up	Rocky	Balaenopteridae		
Jun 24, 2015	Seno Newman	250	46°28.645′S	74°54.83′W			Lateral-up	Rocky	Balaenopteridae		
Jun 24, 2015	Seno Newman	251	46°28.637′S	74°54.831′W	1	1	Lateral-up	Rocky	Balaenopteridae		
Jun 24, 2015	Seno Newman	252	46°28.521′S	74°54.913′W	1	1	Lateral-up	Sandy	Balaenopteridae		
Jun 24, 2015	Seno Newman	253	46°27.411′S	74°54.979′W	1	1	Ventral-up	Rocky			
Jun 24, 2015	Seno Newman	254	46°27.365′S	74°54.984′W	1	1	Lateral-up	Rocky	Balaenopteridae		
Jun 24, 2015	Seno Newman	255	46°27.314′S	74°54.988′W	1	1	Lateral-up	Rocky	Balaenopteridae		
Jun 24, 2015	Seno Newman	256	46°27.214′S	74°54.829′W				Sandy–rocky	Balaenopteridae		
Jun 24, 2015	Seno Newman	257	46°26.271′S	74°53.366′W			Lateral-up	Sandy	Balaenopteridae		
Jun 24, 2015	Seno Newman	258	46°26.119′S	74°53.609′W				Sandy			
Jun 24, 2015	Seno Newman	259	46°26.111′S	74°53.714′W			Lateral-up	Sandy	Balaenopteridae		
Jun 24, 2015	Seno Newman	260	46°26.123′S	74°53.747′W			Lateral-up	Floating	Balaenopteridae		
Jun 24, 2015	Seno Newman	261	46°26.116′S	74°53.771′W			Lateral-up	Floating	Balaenopteridae		
Jun 24, 2015	Seno Newman	262	46°26.264′S	74°54.143′W			Lateral-up	Sandy	Balaenopteridae		
Jun 24, 2015	Seno Newman	263	46°26.336′S	74°54.127′W	1	1	Lateral-up	Sandy	Balaenopteridae		
Jun 24, 2015	Seno Newman	264	46°26.352′S	74°54.148′W	1	1	Lateral-up	Sandy	Balaenopteridae		
Jun 24, 2015	Seno Newman	265	46°26.34′S	74°54.321′W	1	1	Lateral-up	Sandy	Balaenopteridae		
Jun 24, 2015	Seno Newman	266	46°26.335′S	74°54.394′W	1	1	Lateral-up	Sandy	Balaenopteridae		
Jun 24, 2015	Seno Newman	267	46°26.656′S	74°55.481′W	2	1		Rocky			
Jun 24, 2015	Seno Newman	268	46°26.797′S	74°55.902′W				Rocky			
Jun 24, 2015	Seno Newman	269	46°27.022′S	74°56.047′W	1	1		Rocky	Balaenopteridae		
Jun 24, 2015	Seno Newman	270	46°27.248′S	74°56.114′W				Rocky			
Jun 24, 2015	Seno Newman	271	46°27.959′S	74°56.175′W				Sandy–rocky			
Jun 24, 2015	Seno Newman	272	46°28.193′S	74°56.104′W	1	1		Sandy	Balaenopteridae		
Jun 24, 2015	Seno Newman	273	46°28.253′S	74°56.094′W	1	1		Sandy	Balaenopteridae		
Jun 24, 2015	Seno Newman	274	46°28.385′S	74°56.166′W	1	1		Sandy	Balaenopteridae		
Jun 24, 2015	Seno Newman	275	46°28.405′S	74°56.161′W	1	1		Sandy	Balaenopteridae		
Jun 24, 2015	Seno Newman	276	46°28.461′S	74°56.144′W				Sandy–rocky			
Jun 24, 2015	Seno Newman	277	46°29.752′S	74°57.068′W	1	1	Lateral-up	Sandy	Balaenopteridae		
Jun 24, 2015	Seno Newman	278	46°30.896′S	74°58.426′W	1	1	Lateral-up	Sandy	Balaenopteridae		
Jun 24, 2015	Seno Newman	279	46°30.918′S	74°58.439′W	1	1	Lateral-up	Sandy	Balaenopteridae		
Jun 24, 2015	Seno Newman	280	46°31.016′S	74°58.904′W				Rocky			
Jun 24, 2015	Seno Newman	281	46°31.284′S	74°59.402′W	1	1		Sandy–rocky	Balaenopteridae		
Jun 24, 2015	Seno Newman	282	46°31.967′S	74°59.824′W	1	1	Lateral-up	Sandy–rocky	Balaenopteridae		
Jun 24, 2015	Seno Newman	283	46°31.979′S	74°59.845′W	1	1	Lateral-up	Sandy–rocky	Balaenopteridae		
Jun 24, 2015	Seno Newman	284	46°32.007′S	74°59.867′W	1	1	Ventral-up	Sandy–rocky	Balaenopteridae		
Jun 24, 2015	Seno Newman	285	46°31.638′S	75°0.132′W				Rocky			
Jun 24, 2015	Seno Newman	286	46°31.532′S	75°0.959′W				Rocky			
Jun 24, 2015	Seno Newman	287	46°31.767′S	75°0.989′W				Rocky			
Jun 24, 2015	Seno Newman	288	46°31.798′S	75°1.062′W				Rocky			
Jun 24, 2015	Seno Newman	289	46°32.125′S	75°0.925′W	1	1		Sandy–rocky	Balaenopteridae		
Jun 24, 2015	Seno Newman	290	46°32.493′S	75°1.119′W	1	1		Sandy	Balaenopteridae		
Jun 24, 2015	Seno Newman	291	46°32.689′S	75°1.12′W	1		Lateral-up	Sandy	Balaenopteridae		
Jun 24, 2015	Seno Newman	292	46°33.363′S	75°1.351′W	2	1	Lateral-up	Sandy	Balaenopteridae		
Jun 24, 2015	Seno Newman	293	46°33.372′S	75°1.344′W	1	1	Lateral-up	Sandy	Balaenopteridae		
Jun 24, 2015	Seno Newman	294	46°33.428′S	75°1.334′W	1			Rocky	Balaenopteridae		
Jun 24, 2015	Seno Newman	295	46°33.958′S	75°1.688′W				Sandy–rocky	Balaenopteridae		
Jun 24, 2015	Seno Newman	296	46°33.966′S	75°1.732′W	1	1		Sandy–rocky			
Jun 24, 2015	Seno Newman	297	46°33.977′S	75°1.746′W	2	1		Floating			
Jun 24, 2015	Seno Newman	298	46°34.271′S	75°1.855′W	2	1		Rocky			
Jun 24, 2015	Seno Newman	299	46°34.429′S	75°2.047′W	2			Rocky			
Jun 24, 2015	Seno Newman	300	46°34.463′S	75°2.194′W				Sandy–rocky			
Jun 24, 2015	Seno Newman	301	46°38.102′S	75°8.96′W	1	1		Sandy			
Jun 24, 2015	Seno Newman	302	46°38.089′S	75°9.632′W	1			Sandy–rocky			
Jun 24, 2015	Seno Newman	303	46°39.046′S	75°12.857′W				Sandy–rocky	Balaenopteridae		
Jun 24, 2015	Seno Newman	304	46°39.4′S	75°15.631′W				Rocky			
Jun 24, 2015	Seno Newman	305	46°42.092′S	75°14.267′W	1	1		Sandy–rocky	Balaenopteridae		
Other sources	
Middle of March	Bahía Conos	306	46°36.2′S	75°28.7′W	3						
Middle of March	Bahía Conos	307	46°36.229′S	75°28.664′W	3						
Feb 21, 2015	Isla Crosslet	308	46°43.494′S	75°10.521′W	3						
Feb 22, 2015	Isla Crosslet	309	46°45.32′S	75°11.175′W	1				Balaenopteridae		
End of Feb 2015	Fiordo San Pablo	310	46°36.677′S	75°9.685′W	1				Balaenopteridae		
End of Feb 2015	Fiordo San Pablo	311	46°36.271′S	75°9.471′W	3						
End of Feb 2015	Estero Slight	312	46°43.26′S	75°9.37′W	1		Lateral-up	Floating	Balaenoptera borealis	Female	
End of Feb 2015	Estero Slight	313	46°43.26′S	75°9.37′W	1				Balaenopteridae		
End of Feb 2015	Estero Slight	314	46°43.26′S	75°9.37′W	3						
End of Feb 2015	Estero Slight	315	46°47.18′S	75°32.417′W	3						
Middle of Mar 2015	Bahía Conos	316	46°37.007′S	75°27.578′W	1				Balaenopteridae		
Middle of Mar 2015	Bahía Conos	317	46°37.084′S	75°27.664′W	1				Balaenopteridae		
Middle of Mar 2015	Bahía Conos	318	46°37.011′S	75°27.788′W	1				Balaenopteridae		
Middle of Mar 2015	Bahía Conos	319	46°36.918′S	75°27.726′W	1				Balaenopteridae		
Middle of Mar 2015	Bahía Conos	320	46°36.893′S	75°27.881′W	1				Balaenopteridae		
Middle of Mar 2015	Canal Barros Luco	321	50°9.450′S	75°17.317′W	1				Balaenopteridae		
Middle of Mar 2015	Canal Ladrillero	322	49°8.000′S	75°17.000′W	1				Balaenopteridae		
Middle of Mar 2015	South from Isla Solar	323	50°58.975′S	75°4.276′W	1				Balaenopteridae		
Mar 23, 2015	Near Cape Stokes	324	46°54.558′S	75°14.109′W	1			Rocky	Balaenopteridae		
Mar 23, 2015	Near Cape Stokes	325	46°55.76′S	75°16.796′W	1			Sandy	Balaenopteridae		
Mar 23, 2015	Brazo Oeste–Barroso	326	46°50.91′S	75°15.332′W	1			Sandy	Balaenopteridae		
Mar 25, 2015	Brazo Este–Barroso	327	46°51.761′S	75°15.577′W	1				Balaenopteridae		
Mar 5, 2015	Isla Hereford	328	46°43.26′S	75°9.37′W	1				Balaenopteridae		
Mar 5, 2015	Isla Hereford	329	46°43.26′S	75°9.37′W	1				Balaenopteridae		
Mar 5, 2015	Isla Hereford	330	46°43.26′S	75°9.37′W	3						
Mar 5, 2015	Isla Hereford	331	46°35.925′S	75°11.636′W	2						
May 14, 2015	Paso Isaza	332	50°53.983′S	74°18.133′W	1		Lateral-up	Floating	Balaenoptera borealis	Male	
Jul 5, 2015	Near Puerto Natales	333	49°35.733′S	74°26.083′W	1		Lateral-up	Floating	Balaenoptera borealis	Female	
Middle of May 2015	Near Puerto Natales	334	51°28.567′S	73°44.95′W	3						
Middle of May 2015	Near Puerto Natales	335	51°28.392′S	73°44.941′W	3						
Middle of May 2015	Near Puerto Natales	336	51°28.399′S	73°45.399′W	3						
Middle of May 2015	Near Puerto Natales	337	51°28.519′S	73°45.078′W	3						
probably December 2015	Canal Ladrillero	338	49°8.000′S	75°17.000′W							
probably December 2015	Canal Ladrillero	339	49°8.000′S	75°17.000′W							
probably December 2015	Canal Ladrillero	340	49°8.000′S	75°17.000′W							
probably December 2015	Canal Ladrillero	341	49°8.000′S	75°17.000′W							
probably December 2015	Canal Ladrillero	342	49°8.000′S	75°17.000′W							
probably December 2015	Canal Ladrillero	343	49°8.000′S	75°17.000′W							
probably December 2015	Canal Ladrillero	345	49°8.000′S	75°17.000′W							
probably December 2015	Canal Ladrillero	346	49°8.000′S	75°17.000′W							
probably December 2015	Canal Ladrillero	347	49°8.000′S	75°17.000′W							
probably December 2015	Canal Ladrillero	348	49°8.000′S	75°17.000′W							
probably December 2015	Canal Ladrillero	349	49°8.000′S	75°17.000′W							
probably December 2015	Canal Ladrillero	350	49°8.000′S	75°17.000′W							
probably December 2015	Canal Ladrillero	351	49°8.000′S	75°17.000′W							
probably December 2015	Canal Ladrillero	352	49°8.000′S	75°17.000′W							
probably December 2015	Canal Ladrillero	353	49°8.000′S	75°17.000′W							
probably December 2015	Canal Ladrillero	354	49°8.000′S	75°17.000′W							
probably December 2015	Canal Ladrillero	355	49°8.000′S	75°17.000′W							
probably December 2015	Canal Ladrillero	356	49°8.000′S	75°17.000′W							
probably December 2015	Canal Ladrillero	357	49°8.000′S	75°17.000′W							
probably December 2015	Canal Ladrillero	358	49°8.000′S	75°17.000′W							
probably December 2015	Canal Ladrillero	359	49°8.000′S	75°17.000′W							
probably December 2015	Canal Ladrillero	360	49°8.000′S	75°17.000′W							
probably December 2015	Canal Ladrillero	361	49°8.000′S	75°17.000′W							
probably December 2015	Canal Ladrillero	362	49°8.000′S	75°17.000′W							
probably December 2015	Canal Ladrillero	363	49°8.000′S	75°17.000′W							
probably December 2015	Canal Ladrillero	364	49°8.000′S	75°17.000′W							
probably December 2015	Canal Ladrillero	365	49°8.000′S	75°17.000′W							
probably December 2015	Canal Ladrillero	366	49°8.000′S	75°17.000′W							
probably December 2015	Canal Ladrillero	367	49°8.000′S	75°17.000′W							
HF26 expedition	
Feb 4, 2016	Bayron	368	47°48.102′ S	74°58.235′W	1	1		Floating	Balaenopteridae		
Feb 6, 2016	Seno Escondido	369	46°50.885′S	74°27.675′W	1	1		Sandy–rocky	Balaenopteridae		
Feb 13, 2016	Seno Slight	370	46°42.880′S	75°28.803′W	1	1		Sandy–rocky	Balaenopteridae		
Feb 14, 2016	Seno Slight	371	46°48.525′S	75°34.157′W	1	1		Sandy–rocky	Balaenopteridae		
Feb 14, 2016	Seno Slight	372	46°47.800′S	75°32.773′W	1	1		Sandy–rocky	Balaenopteridae		
Feb 15, 2016	Seno Slight	373	46°47.272′S	75°29.853′W	1	1		Sandy–rocky	Balaenopteridae		
Feb 15, 2016	Seno Slight	374	46°46.232′S	75°31.137′W	1	1		Sandy–rocky	Balaenopteridae		
Feb 18, 2016	Newman	375	46°29.557′S	74°55.182′W	1	1		Sandy–rocky	Balaenopteridae		
Feb 22, 2016	Newman	376	46°30.672′S	74°55.607′W	1	1		Sandy–rocky	Balaenopteridae		
Feb 23, 2016	Caleta Buena	377	46°47.072′S	75°29.847′W	1	1		Sandy–rocky	Balaenopteridae		
Feb 23, 2016	Caleta Buena	378	46°47.233′S	75°29.843′W	1	1		Sandy–rocky	Balaenopteridae		
Feb 24, 2016	Slight	379	46°47.233′S	75°29.843′W	1	1		Sandy–rocky	Balaenopteridae		
Feb 24, 2016	Slight	380	46°48.413′S	75°34.772′W	1	1		Sandy–rocky	Balaenopteridae		
May 2016	Seno Escondido	381	46°49.963′S	74°39.016′W				Floating			
May 2016	Slight	382	46°47.444′S	74°34.460′W							
May 2016	Newman	383	46°30.672′S	74°55.607′W							
Other sources	
Feb 6, 2016	Islas Jungfrauen	384	47°55.527′S	75°6.832′W							
Mar 13, 2016	Ushuaia	385	54°53.756′S	67°22.571′W	1	1		Floating	Megaptera novaeangliae		
Mar 28, 2016	Navarino	386	54°55.350′S	68°18.555′W	2	1			Megaptera novaeangliae		
Jan 2016	Canal Ladrillero	387	49°8.000′S	75°17.000′W				Floating			
Jan 2016	Canal Ladrillero	388	49°8.000′S	75°17.000′W				Floating			

Figure 4 Documented whale carcasses and skeletal remains during an overflight on Jun 25, 2015, Seno Escondido.

The numbers correspond to the whale identification numbers in Table 1. Photos: Verena Häussermann, all rights reserved.

In addition to the whale carcasses and skeletons from the two surveys, some whale carcasses and skulls were reported between Feb and Jun 2015 by boat crews navigating the west coast of Taitao Peninsula and the coast between 49°15′ and 51°S (Table 2). Between Jan 23 and Mar 1, 2016 (Expedition Huinay Fiordos 27) and between Apr 27 and May 30, 2016 (Expedition Huinay Fiordos 29), two additional vessel-based expeditions were carried out, each to Seno Escondido, Seno Newman and Estero Slight, with the aim of searching for new carcasses, taking samples for genetic and red tide analyses, and performing oceanographic transects. Data from those surveys are included here, but most of the analyses of the samples will be published in a separate paper.

Samples of marine invertebrates were collected under permit of Subsecretaria de Pesca y Acuicultura (R.EX. 1295 del 27.04.2016). Samples of cetacean carcasses were authorized by SERNAPESCA, Region de Aysen (Acta Numbers 2016-11-10 and 12).

Satellite image

A high-resolution satellite image was taken of Seno Newman on Aug 13, 2015 using the Pleiades-1 Satellite. The 16-bit ortho-rectified GeoTIFF multispectral (R-G-B-NIR) and Panchromatic files have been analyzed to count whale carcasses and determine their geographic positions (Fig. 5). The whales identified in the satellite image were compared to the photos and GPS locations obtained during the overflight, and cross-matched with reference to nearby geomorphological features.

Figure 5 (A) Satellite image on Aug 13, 2015, used to count the carcasses along Seno Newman.

(B–D) Detail of the carcasses highlighted in (A).

Taxonomic analysis

Whales were identified in situ during the vessel-based expedition based on morphological characteristics. The species identification of the specimens from which tissue was sampled during the SERNAPESCA expedition to Estero Slight was confirmed genetically (Fiscalía de Aysén, 2015). A 675 bp fragment of mitochondrial DNA control region was amplified using the primers using the primers M13 Dlp1.5 5′-TGTAAAACGACAGCCAGTTCACCCAAAGCTGRARTTCTA-3′ and 8G 5′GGAGTACTATGTCCTGTAACCA (Dalebout et al., 2005) and sequenced in both directions. Amplification reactions were performed in a total volume of 25 μl with 5 μl PCR buffer 10×, 2 μl MgCl2 50 mm, 1 μl of each primer, 2 μl dNTP 200 mm and 0.3 μl Taq DNA polymerase (Invitrogen Life Technologies, Carlsbad, CA, USA) and 50 ng DNA. The PCR temperature profile was as follows: a preliminary denaturing period of 2 min at 94 °C followed by 30 cycles of denaturation for 30 s at 94 °C, primer annealing for 40 s at 56 °C and polymerase extension for 40 s at 72 °C. A final extension period for 10 min at 72 °C was included.

Taphonomy

Analysis was carried out, following biostratinomic criteria, on different subsets of the whale remains recorded during the overflight and the vessel-based surveys. Characterization of the depositional state of the carcasses was based on a post hoc analysis of the assemblage, exclusively through photographs, classifying the carcasses into three taphonomic classes according to previous studies of biostratinomic processes in marine mammals (Pyenson et al., 2014, Liebig, Taylor & Flessa, 2003; Liebig, Flessa & Taylor, 2007; Schäfer, 1972). The aspects considered were anatomic position of the carcasses (ventral, dorsal or lateral side-up, n = 201), deposition site (rocky or sandy, n = 295), and the disarticulation and degree of decay of the carcasses. These final two aspects were sorted into classes to estimate the sequence of disarticulation/decay addressing two aspects: time since death (n = 245) and drift time/distance of the carcass (as a proxy to estimate the relative location of death, n = 151).

To assess the time since death, three categories were defined, reflecting a straightforward order from the least decomposed to the most disarticulated carcass/skeleton. “Class 1” refers to carcasses in the lowest to relatively medium state of decomposition for these assemblages. Included in this category are complete carcasses with skin, complete carcasses without skin, and complete carcasses with partially exposed bones (see Fig. 6A). “Class 2” includes carcasses in a relatively greater state of decomposition but still maintaining their longitudinal axis, although some bones may be scattered (see Fig. 6B). Finally, “Class 3” refers to isolated skeletal remains with no soft tissue, such as skulls, dentaries or postcranial remains (see Fig. 6C). Thus, the sequence of “time since death” should reflect ranges from less than three months (Class 1), several months, but probably less than six months (Class 2), to a year or more (Class 3).

Figure 6 Biostratinomic classification addressing the decomposition/disarticulation of carcasses/skeletal remains assessing to the time since death.

(A and B) Class 1, carcasses in the lowest to relatively medium state of decomposition. (C and D) Class 2, carcasses in a relatively greater state of decomposition, but still maintaining their longitudinal axis, although some bones may be scattered. (E and F) Class 3, isolated skeletal remains with no soft tissue. Photos: Verena Häussermann (A–D), Photos: Ana Valenzuela-Toro (E, F), all rights reserved.

The analysis of the location of death, namely whether the carcasses are para-autochthonous or allochthonous was addressed by evaluation of the time that the carcasses had remained floating in the water column and at the surface (see Schäfer, 1972). For this, we defined two classes, depending of the presence or absence of the skull, as a proxy for the time floating and the potential distance between the site of mortality and the observed site of deposition (Fig. 7) (Toots, 1965; Voorhies, 1969; Behrensmeyer, 1973; Holz & Simões, 2002; Liebig, Taylor & Flessa, 2003; Simões & Holz, 2004). Thus, “Class A” includes carcasses that preserve the skull and “Class B” includes those without a skull. For this analysis, we excluded skeletons, which were considered older than a year (minimum age, based on field observations of AVT from 2016 expedition to the site of the mortality).

Figure 7 Biostratonomic classification of the location of death of carcasses/skeletal remains.

(A) Carcasses preserving the skull. (B) Carcasses lacking the skull. Photos: Verena Häussermann (A), Fanny Horwitz (B), all rights reserved.

A geomorphological analysis was made using photographs and Google Earth (Terrametrics, 2015). We classified the type of depositional locality (i.e., sand/pebble dominated beach or rocky outcrop) (Table 2) in order to assess the relationship between these aspects and the taphonomic categories mentioned above; for instance, whether carcasses that had been transported further and disarticulated (allochthonous) were more prevalent at high energy sites (i.e., rocky outcrops) and articulated (para-autochthonous) carcasses more prevalent in low energy environments (i.e., sandy beaches).

To compare the density of the death assemblages at Golfo de Penas with known extinct and extant death assemblages recorded in the literature, we measured linear dimensions of the geomorphological units (i.e., length and width of the beach), through the measure tool in Google Earth, using the highest resolution satellite images available, at sites where assemblages were found. In this manner, the geographic areas corresponding to the death assemblages were calculated and the density determined by dividing the number of specimens in each assemblage by its area.

Analysis of the petrotympanic complex (ear bone)

We studied the bones of the middle and inner ear of one whale, collected during the SERNAPESCA expedition. A volumetric computed tomography in the Morita tomography (box of 60 mm, 500 cuts) was carried out. The images were visualized with Osirix Dicom viewer v 5.6 32-bit in search for fractures or micro-fractures, which would appear as black gaps in the bony tissue.

Analysis for toxins (PST/AST)

Bivalve tissue was sampled in Estero Slight on Apr 22 and on May 25, 2015 (two samples in total), and in Estero Slight, Seno Newman and Seno Escondido between Jan 23 and Mar 1, and Apr 27 and May 30, 2016 (22 samples in total). The stomach content and intestine content of two whales from Estero Slight were sampled on May 25, 2015. On Feb 2016, one sample of duodenum content was obtained from a freshly dead whale observed in Estero Slight. At the same period, one sample of surface-swimming Munida spp. was collected at 46°29.730′S, 74°55.722′W. All samples were analyzed in situ for presence of PST using the protocol already described for the shellfish tissue and stomach content samples. The tissue was homogenized using a blender and mixed in a 1:1 ratio with a field extraction fluid composed of 2.5 parts of rubbing alcohol (70%) to one part white vinegar. The mixture was then homogenized manually and filtered through a paper filter (paper filter #4). The extract obtained after filtration was then used to detect the presence of toxins through rapid field test kits from scotia rapid testing for PST and AST. For this, 100 μl of the extract was placed in a test tube containing running buffer, mixed and then 100 μl of this mixture was placed in a lateral flow enzyme-linked immunosorbent assay (ELISA) test strip with antibodies specific for PST (saxitoxin and its derivative toxins) and AST (domoic acid). These tests were left to develop for 1 h before the results were read.

Twenty-two phytoplankton samples were collected in Estero Slight, Seno Newman and Seno Escondido between Jan 23 and Mar 1, and Apr 27 and May 30, 2016, using a 20 μm mesh size plankton net in a vertical tow from 15 m depth. The phytoplankton present in these samples was concentrated using the net, and a 100 μl subsample was placed in a tube with 0.1M acetic acid and mixed. About 100 μl of this mixture were then added to a test tube-containing running buffer and an aliquot of this mixture of the same volume was placed in an ELISA test strip for PST and left to develop for 1 h before results were read.

These qualitative PST test strips are extremely sensitive due to the local toxin profile, which is high in GTX2/3, resulting in detection limits below 32 μg STX Eq/100 g of tissue. The detection limit for the AST tests was reduced to 2 ppm of domoic acid by modifying the standard sample preparation protocol by eliminating the dilution of the sample before mixing it with the buffer.

A graphical analysis of the geographic and temporal distribution of PSP events, presence of harmful microalgae and environmental variables in the affected region (43°S–51°S) from 2007 to Jul 2015 and from Mar 2016 was performed with the data obtained from the red tide monitoring program conducted by the SERNAPESCA (R.S. Galdames, 2015, personal communication), in which mytilid samples are analyzed at several stations throughout Chilean Patagonia approximately once a month by the “Laboratorios SEREMI Salud,” from Aysén and Magallanes regions at Southern Chile.

Drift model

Floating objects are directly affected by surface currents, wind and waves. Wind both drives the Ekman drift of surface water (Ardhuin et al., 2009) and exerts a direct drag on the emerged surface of an object (Breivik et al., 2012). Stokes drift, the net forward transport due to non-closed particle trajectories resulting from passing waves, also contributes to the transport of floating objects. The drift of whale carcasses was simulated by parameterizing the contribution of these components, based on objects of a similar size from search and rescue models (Breivik et al., 2012; Peltier et al., 2012). Due to the large uncertainty in carcass drift characteristics, parameters were varied stochastically within a wide range of possible values.

Use was made of existing current and wave products, the HYCOM daily 1/12 simulation (Wallcraft, Metzger & Carroll, 2009), and waves from ECMWF ERA-Interim reanalysis (Dee et al., 2011). Winds were taken from a custom downscaling of NCEP NFL boundary conditions using the WRF model (Skamarock & Klemp, 2008) to a sub-4 km grid size. Drift scenarios were run by stepping forward in time from hypothetical sites and times of mortality. All of these sites were in shallow water, since carcasses resulting from mortality in deep water have a tendency to sink and not resurface (Smith et al., 2015). A horizontal diffusion coefficient of 10 m2s−1 was included in drift tracks to represent unresolved physical processes. While the resolution of the current and wave datasets is inadequate to represent detailed coastline or seabed geometry, or the interior of the fjords, the drift model does clarify the expected distribution and spread of carcasses from localized sources.

Large-scale wind stress

The large-scale tendency toward upwelling or downwelling provides a key driver of coastal ecosystems. This was assessed using ECMWF ERA-Interim reanalysis data (Dee et al., 2011). It is the alongshore component of wind stress that drives Ekman transport normal to the coast and consequent upwelling or downwelling. Since upwelling and downwelling are cumulative processes, a time-integrated wind stress was calculated (Pierce et al., 2006) from a base time of the vernal equinox (September 21). Stress was estimated from reanalysis winds at 10 m elevation according to Large & Pond (1981). The large-scale change in coastal orientation was taken into account in extracting the alongshore wind component, although localized inlets, bays (including the Golfo de Penas) and islands were not considered.

Results

Field surveys and toxicity tests

Of the total of dead whales observed in all expeditions and reports in 2015 (367), 35 recently dead whales and 12 skeletal remains were discovered during the HF24 expedition: 31 carcasses and 12 skeletal remains were found in and close to the entrance of the 14 km long Estero Slight and four carcasses in Canal Castillo, situated 235 km to the south, as well as many whale bones on different beaches (Fig. 3; Table 2). Three hundred and five carcasses were mapped during the overflight between the Jungfrauen Islands (∼48°S) and Seno Newman (46°39′S). In addition to this total of 284 whale carcasses and 21 skeletons from the two surveys, 51 whale carcasses and 11 whale skulls were reported between Feb and Jun 2015 by boat crews navigating the west coast of Taitao Peninsula and the coast between 49°15′ and 51°S (Table 2; Fig. 4).

On some photos what could have been carcasses of smaller animals (possibly dolphins and/or sea lions) were seen, but due to the flying altitude, speed and weather conditions, the photo quality and resolution did not allow their conclusive identification as actual carcasses. In Estero Slight, one dead pinniped was found on the shore from the vessel. During the SERNAPESCA expedition, one Otariidae skull was found and photographed in the same channel but the correspondence of the carcass and the skull could not be established.

The 28 whale carcasses that could be identified unambiguously to species level were all sei whales (Balaenoptera borealis); 15 of these identifications were confirmed genetically. Seven specimens could be identified as males and ten as females. One hundred and twenty-nine carcasses were identified as baleen whales of the Balaenopteridae family or rorquals. The 30 whales examined in detail in Estero Slight during the vessel-based expedition were between 6 and 15 m long, hence included both juvenile and fully grown specimens.

None of the examined whales showed any evidence of disease or traumatic damage. The anatomic structures of the ear bone were in good condition showing no damage; the stapes were articulated in place, and the bony tissue showed no fractures (Fig. 8). The analysis of locally collected mytilids in Apr and May 2015 and of the stomach and intestine content of two whales in May 2015 showed presence of PST and AST.

Figure 8 Digital images obtained through computed volumetric tomography (CVT) scanned at Morita tomography (box of 60 mm, 500 slices).

All acoustic anatomical structures of the middle ear (ossicles: stapes), internal ear (cochlea: spiral lamina), and the semicircular canals are seen in perfect condition. Transversal sections of the pars cochlearis of the periotic: (A) midline, (B) more anterior; sagittal sections of the pars cochlearis of the periotic, (C) anterior, (D) midline and (E) posterior; Lateromedial sections of the pars cochlearis of the periotic: (F) lateral, (G) half-length and (H) medial.

In 2016, 16 fresh carcasses were observed during the HF27 and HF29 vessel-based expeditions to Golfo Tres Montes; five further were reported by boat crews navigating the Southern part of Chilean Patagonia. None of the examined whales showed any evidence of disease or traumatic damage. Thirty-six rapid tests on PST were run using mussels (12 tests), Munida (two tests), and phytoplankton (22 tests) in Seno Escondido, Seno Newman and Estero Slight. Most of the samples collected during the 2016 expeditions proved to be negative for the presence of PST, nevertheless, both expeditions detected the presence of PSP in the phytoplankton collected at the entrance of Seno Newman. A sample collected at the head of Seno Newman was negative for PST, indicating that the toxic phytoplankton was preferentially located at the mouth of this inlet and nearby areas of the Canal Chaicayán.

Biostratinomic analysis

Of the 367 dead whales observed in 2015, 305 carcasses were mapped between Seno Newman (46°39′S) and Jungfrauen Islands (∼48°S). Those carcasses could be grouped into five assemblages (Figs. 1 and 9; Table 2), defined as a group of carcasses in close proximity. The assemblages were called Golfo de Penas, Jungfrauen Islands, Seno Escondido, Seno Newman and Estero Slight.

Figure 9 Maps showing the five assemblages of whale carcasses.

(A) Golfo de Penas, (B) Seno Escondido, (C) Seno Newman, (D) Estero Slight and (E) Jungfrauen Islands. State of decomposition color-coded: yellow (state 1; least decomposed, all articulated), orange (state 2; intermediate decomposed), and red (state 3; isolated remains).

Some carcasses were floating (11), but most (284) were deposited ashore (Figs. 3–5). The greater proportion of carcasses were deposited in a lateral position and to a lesser extent in the ventral-up position reflecting the hydrodynamics of the body in the sea as determined by the inflation of the abdominal region and mainly of their tongues, as observed in a recently dead individual and in some decayed carcasses at Golfo de Penas (Fig. 10). In general, they were tide-oriented (parallel to the coastline) and all of the classified carcasses from the overflight were lying on their back or side (ventral-up, 44.3%; lateral-up, 55.7%) (Table 3; Fig. 11C), while only one specimen (from HF24) was found in a dorsal-up position (data not included in analysis due to different time of observation).

Figure 10 Inflation of the tongue and its implication for whale carcass deposition.

(A) Inflated tongue in a very recently dead sei whale (weeks) indicated by the arrowhead. (B) Close-up of the mouth with dislocate mandibles due to the previous inflation of the tongue (arrowhead), which is decayed and removed by scavengers. (C) Whale carcass seen from the overflight deposited in lateral position and its protuberant inflated tongue (arrowhead). Photos: Brice Monégier (A), Verena Häussermann (B, C), all rights reserved.

Table 3 Anatomical position.

Proportion of carcasses in each anatomical position as recorded from the overflight survey and posterior photographic analysis.

Anatomical position of Carcass	Unknown	Dorsal-up	Ventral-up	Lateral-up	Total	
Count	187	0	43	54	97	
Proportion (%)	65.84	0	15.14	19.01	100	
Proportion (%) based on classified individuals only	–	0	44	56	100	

Figure 11 Graphs showing the proportion of the total classified carcasses in the biostratonomic analysis.

(A) Time since death. (B) Time since death, combining Class 1 and 2. (C) Location of death and (D) Anatomical positions of carcasses (lateral, ventral and dorsal-up).

With respect to the classification of “time since death,” 68.8% of the carcasses were classified in Class 1 (less than three months), 24.9% in Class 2 (less than six months) and 6.3% in Class 3 (more than a year) (Figs. 11A and 11B; Table 4). With respect to “time at sea,” 147 (87%) of the carcasses were classified in Class A (short time/distance of drift), while only four (13%) were identified as Class B (long time/distance of drift) (Fig. 11C; Table 4). There was no pattern relating the geomorphological unit (sandy: 34%, pebble: 27%, rocky beach: 34%) to the taphonomic classes.

Table 4 Minimal number of individuals (MNI).

Estimation of minimal number of individuals are given to each of the classes of decomposition/disarticulation stages recorded at Golfo de Penas.

Classes of decomposition	Class	MNI	Proportion (%)	
Time since death	1	141	68.78	
2	51	24.88	
3	13	6.34	
Total	205	100	
Time at sea	A	147	97.35	
B	4	2.64	
Total	151	100	

The carcasses found in April in Estero Slight were classified in stage 2 of Geraci & Lounsbury (2005) indicating a few days to weeks since death; this would be classified as Class 1 in the taphonomic classes of the present study.

The density of whale carcasses was in average 1,050/km2, considering all assemblages recognized (Table 5).

Table 5 Density of specimens in assemblages (specimens/km2).

	Area (km2)	Number of specimens	Density (specimens/km2)	
Assemblage 1—Jungfrauen group	0.19	30	156	
Assemblage 2—Escondido inlet	0.02	47	1,906	
Assemblage 3—Escondido inlet	0.01	32	1,987	
Assemblage 4—Newman inlet	0.60	149	248	
Assemblage 5—Slight inlet	0.04	40	952	
Total area of assemblages/specimens	0.87	298	341	
Average	0.17	59	1,050	

Carcass drift and potential source locations

The distribution of beached carcasses was simulated from four illustrative source locations (Figs. 12A–12D). In each case, calculations tracked 13,000 hypothetical carcasses, reflecting source times spanning a two-month period from mid-February to mid-April 2015 and a range of drift model parameters. The spread of stranding locations therefore represents variability of the current, wind and wave environment during this period as well as the uncertainty in model parameters and a diffusive component to the drift tracks. While each of the illustrated source locations leads to strandings distributed over several hundred kilometers of coastline, there are important differences in these distributions. A simulated source in Golfo Tres Montes (Northern Golfo de Penas) leads to strandings throughout the Golfo de Penas (Fig. 12A), including in the Golfo Tres Montes itself. No other source location (Figs. 12B–12D) leads to strandings in Golfo Tres Montes due to the direction of prevailing currents and the sheltering effect of Peninsula Taitao. Similarly, only a source to the north of Peninsula Taitao leads to strandings in that region (Fig. 12B). Carcasses originating in the Golfo de Penas have a tendency to be transported to the south by prevailing currents (Figs. 12A, 12C and 12D).

Figure 12 Location of beached carcasses (blue) predicted by the drift model from four possible mortality locations (A–D, red stars).

Mortalities during a two month period are simulated, from mid-February to mid-April 2015, with multiple carcasses (n = 200) of varying drift properties released each day to predict the range of resulting carcass locations. Green vectors show time-averaged surface currents for this period (HYCOM model). Depth contours at 50 and 100 m are indicated (GEBCO), although nearshore waters and inlets are not resolved.

Inter-annual variation in upwelling or downwelling

Comparison between the cumulative alongshore wind stress for the year in question and the previous 20 years (Fig. 13) reveals that the months immediately prior to the mortality event were anomalous. North of the study area, at 45°S, there was an anomalously strong tendency toward upwelling (an upward trend in Fig. 13), making this one of the most upwelled years of the period. At the latitude of Golfo de Penas and further south there was a net tendency to downwelling (a downward trend in Fig. 13), but punctuated by upwelling events, making this one of the least downwelled years of the period.

Figure 13 Cumulative alongshore component of nearshore wind stress (red) from ECMWF ERAInterim reanalysis winds at latitudes (A) 49°S, (B) 47°S, (C) 45°S, with an origin time of the vernal equinox, Sep 21, 2014.

Gray shading shows the envelope of variability experienced during 1995–2014, with darker shading indicating one standard deviation from the mean for this period. Vertical lines show the timing of vessel (green) and aerial (blue) observations of whale carcasse.

Discussion

Possible causes of death (Table 6) need to be analyzed for a mechanism that is capable of synchronous killing of hundreds of whales, apparently all or most of the same species (with a few exceptions, i.e., one confirmed pinniped). Baleen whales, in contrast to odontocetes, are less social and do not use echolocation to navigate (Perrin, Mead & Brownell, 2009). The latter characteristics are key aspects used to explain mass mortalities in odontocetes.

Table 6 Comparison of the usual causes of death with the evidence encountered at Golfo the Penas.

Cause of death for marine mammals	Main feature	Type of evidence (confirm–discard)	Observation at Golfo de Penas	Expected in rorqual event	Oceanographic conditions near time of death	Rorqual species recorded	References	Oceanographic conditions observed at GP	
Starvation by abundance surpassing carrying capacity	Thin blubber layer, or empty stomach, or numbers around 5% of population	Measurements, necropsy and population numbers nearby carrying capacity	Not likely, sei whales are still recovering from whaling, last species to be hunted	Reported in one species	Low productivity event	Reported in gray whales (Eschrichthius robustus)	Gulland et al. (2005)	High productivity event	
Epidemic disease	Morbillivirus: contagious-epidemic, emaciated, external and internal parasites, lesions and inflammatory reactions	Histology, parasitology–virology test	No signs of external or internal lesions in the whales of Estero Slight
Stomach content present
No test available	Low numbers, young individuals	Shift in temperature, anthropogenic contamination, mutation of virus	Juveniles and calves fin whales,
Balaenoptera physalus	Brongersma-Sanders (1957), Jauniaux et al. (2000), Shimizu et al. (2013), Van Bressem et al. (2014), Mazzariol et al. (2016)	Unknown	
Military exercise with sonar	Only confirmed in dolphins	Ear damage and—or hemorrhage nearby the ears	Unknown	Unknown	No military exercises public programed,
Chilean law for the protection of whales	Unknown	Goldbogen et al. (2013), Nowacek et al. (2007), Southall et al. (2009)	Not reported	
Poisoning by toxins of harmful algal bloom	Massive, multispecific, recurrent in time	HAB reported, shift in oceanographic conditions, El Niño event	Yes	Yes	High productivity event, El Niño influence	Balaenoptera physalus, Megaptera novaeangliae, Balaenoptera acutorostrata	Geraci et al. (1989), Fire et al. (2010), Pyenson et al. (2014), Brongersma-Sanders (1957) (present work)	Yes, at the closest station of red tide monitoring	
Trauma: ship collision/entanglement	Evidence of trauma, small number (i.e., eight deaths in 19 years in USA)	Lesions, hematoma	No sign of internal or external lesion	Yes (small number of individuals at a time)	Not related	Eubalaena glacialis	Kraus (1990), Moore et al. (2004), Vanderlann & Taggart (2007)	Not related	

Possible causes for the death of hundreds of baleen whales include a lethal and highly contagious unknown virus or infection, noise-related mechanisms at sea, and intoxication by biotoxins (domic acid, saxitocin, etc.; Geraci et al., 1989; Fire et al., 2010; Lefebvre et al., 2016; Pyenson et al., 2014; Table 6). In this assemblage, the individuals could not be tested for viruses or bacteria, due to their advanced state of decomposition. There was no evidence of pathological modifications that could be attributed to such a cause; however, it is not possible to completely discard this hypothesis.

The only potentially lethal noise-related mechanism for a baleen whale are very intense noises associated with blasting in close proximity (Ketten, 1992). This could injure the animal and cause hemorrhage or provoke panic, disorientation and favor entrapment (not yet described for baleen whales, Goldbogen et al., 2013). Although there was no evidence of bony damage or micro-fracture of the one examined periotic, this cannot be excluded for the other individuals. Any other noise-related damage could neither be ruled out due to the decomposition of the soft tissue structures, nevertheless, there is no evidence that for baleen sonar and ground noise could trigger more than non-lethal behavioral and temporary effects (Goldbogen et al., 2013). The strongest argument against this hypothesis is that whales died synchronously along hundreds of kilometers of shoreline and at least five different sources of carcasses were identified (see discussion on drift models), which could only be explained by a large number of blastings along the coast during a very restricted time period. The study carried out by SERNAPESCA (Fiscalía de Aysén, 2015; Ulloa et al., 2016, available upon request from SERNAPESCA authorities) based on partial necropsies of two whales in late May 2015, found no evidence of any trauma or human interaction. The whales were already in decomposition stages 3–4 and Class 1 of taphonomic classes used here.

Paralytic shellfish toxin is known to accumulate in the pelagic stage of the squat lobster Munida gregaria (MacKenzie & Harwood, 2014), an important prey of sei whales (Matthews, 1932). Older reports (Tabeta & Kanamura, 1970) and recent observations by boat crews (K.-L. Pashuk, 2015, personal communication) indicate that squat lobster abundance fluctuates strongly and can reach extremely high concentrations, especially in Golfo Tres Montes (Tabeta & Kanamura, 1970). The presence of PST in mytilids from the area and in the whale carcasses and the absence of evidence for other causes of death leaves PSP as the most probable cause of death (Table 6). Although AST was also detected in one of the stomach content samples, it is not believed to be the cause of the MME as it was not detected by the toxin monitoring stations. A mixed assemblage of 40 skeletons from the Miocene in the north of Chile, dominated by rorqual whales and attributed to four recurrent HAB events, shows many similarities to the assemblages described here (Pyenson et al., 2014). The characteristics of the MME and the repetition in the same locality are common features for HAB-mediated mortalities (Brongersma-Sanders, 1957) (see Tables 6 and 7). MMEs through PSP in rorquals are thus not a recent phenomenon in the Southeast Pacific. Nevertheless, whalebone accumulations and reports of mortalities in Chilean Patagonia of up to 15 rorquals going back to at least 1977 suggest an increase in the frequency of mortalities (Table 8). Since the early 1990s, HABs have been recorded every year in spring and autumn along the entire Patagonian coast, patterns are patchy and generally restricted to bays and fjords. The same is true the coast of the Northeast Pacific where HAB events have been increasing in strength and extension (Cook et al., 2015). This MME coincided with increased mortality of baleen whales along the west coast of North America in 2015 (NOAA, 2015b), and with the most extended and longest lasting HAB event registered there (NOAA, 2015c). A positive correlation between the occurrence of PST blooms and the ENSO indices in northern and central Patagonia has been shown (Cassis, Muñoz & Avaria, 2002; Guzmán & Pizarro, 2014). A similar correlation between the abundance of toxic harmful algae and surface temperatures, which in turn are affected by ENSO, was observed in Aysén by Cassis, Muñoz & Avaria (2002). El Niño events have increased in frequency and strength due to global warming (Cai et al., 2014). A strong El Niño event began to build in Sep 2014, which became the strongest El Niño of all time (NOAA, 2015a). The calculated cumulative windstress (Fig. 13) suggests that during this period there was an anomalous tendency toward coastal upwelling and associated nutrient delivery. Exceptionally high levels of PST, 10 times higher than usual peaks were reported in Mar 2015 from the closest monitoring site 120 km north of the mortality area (Isla Canquenes, Fig. 14).

Table 7 Main biostratinomic pathways and their significance in understanding the thanatocenosis.

Time since death	Condition of the carcasses	Age proportions	Sex proportions	Geographic position	Observed	
Catastrophic—single event	Highly homogenous
Majority within one to a few classes (42)	Same as population rate	Same as population rate	Homogenous	Homogenous; see Table S2	
Time averaged	Highly heterogeneous
Several classes present	Same as proportion of annual mortality of the population	No pattern, different from ratio of population	Heterogeneous	Homogenous; see Table S2	
Location of death	Condition of the carcasses	Anatomic position expected	Anatomic position expected	Orientation	Anatomic position observed	
Autochthonous	Very well preserved, low disarticulation	Position of life: dorsal-up (5)	Dorsal-up	No trend	Dorsal-up: 1.00%	
Allochthonous	Disarticulation and scattering present, depending on time and distance to final deposit	Heterogeneous depending on time since death or time of drift
Majority ventral to lateral up (5, see Fig. S5)	Ventral-up—lateral-up	One main direction (current-wind) and/or two main directions (tide)	Ventral-up: 20.40%
Lateral-up: 78.61%	

Table 8 Sei whales observed in Chilean Patagonia (whaling ended in 1976).

Region/site	Number of whales	Time span	Distance to shore (mi)	Source	
43–45°S	286	Mar 25–Apr 03, 1966	60–70	Aguayo-Lobo (1974)	
39–41°S	345	Oct 09–20, 1966	60–120	Aguayo-Lobo (1974)	
46–48°S	114	Dec 13–23, 1966	20–60	Aguayo-Lobo (1974)	
Golfo de Penas (∼46°30′–48°S)	600	Mar 1966	11–24	L. Pastene, 2015, personal communication	
Golfo de Penas (∼46°30′–48°S)	Small number	May 25–28, 1971	Inshore	Gilmore (1971)	
53–55°S	Large concentrations	Feb 1994	Not mentioned	Pastene & Shimada (1999)	
Slight inlet (∼46°45′S)	Two	Jul 2015	Near to shore	J. Cabezas, 2015, personal communication	

Figure 14 Spatial distribution of PST (STX. Eq./100 g tissue) as measured in mytilids and the relative abundance of Alexandrium catenella between 43°S and 51°S in Mar 2015.

Inset shows the toxin level at the closest site to the Golfo de Penas, Isla Canquenes (45°43′31″S; 74°06′51″W) measured between Mar 2010 and Mar 2015. Shellfish consumption is unsafe for humans if values rise above 80 μg STX. Eq./100 g tissue.

The presence of PST during Feb 2016 was accompanied by deep red/brown surface water discoloration due to the high abundance of Alexandrium catenella. This HAB was coincidental with an unusually large bloom of the same toxic species in the waters around Chiloé island (42°S) (Hernández et al., 2016). The May 2016 expedition did not observe water discoloration at this location, nevertheless the phytoplankton samples obtained at the mouth of Seno Newman were also positive for PST, indicating that this toxic species can be present in the area for long periods of time during the summer. The PST levels at Isla Canquenes were not elevated in 2016; however, at two sites in the Messier Channel levels four and seven times higher than usual peaks, were measured (Fig. 15).

Figure 15 Spatial distribution of PST (STX. Eq./100 g tissue) as measured in mytilids between 43°S and 51°S in Mar 2016.

In 2016, the PST levels in the Golfo Tres Montes region were not elevated. However, values four to seven times higher than usual peaks were measured in the channels of Central Patagonia. Shellfish consumption is unsafe for humans if values rise above 80 μg STX. Eq./100 g tissue.

Rorqual whales sink shortly after death (Smith et al., 2015). Once carcasses have sunk below a depth of 50–100 m, they tend not to re-float since hydrostatic pressure compresses decomposition gases (Smith et al., 2015). The bathymetry in the Golfo de Penas area and off the steeply sloping Taitao Peninsula (Fig. 12) requires that the whales that washed ashore all died near the shore. Thus, we conclude that despite common belief (Perrin, Würsig & Thewissen, 2009) sei whales opportunistically feed close to shore and may even follow their prey into narrow and shallow inlets and channels. This hypothesis is supported by the fact that live sei whales were observed near shore in Golfo de Penas and Estero Slight on several occasions (Table 8).

The drift model suggests that the observed carcasses originated from multiple sites. The carcasses found in the two fjordic inlets of Seno Newman and Estero Slight (62% of the total) probably died not far from where they stranded, either in the Golfo Tres Montes or within the inlets themselves (Figs. 1 and 9), since source locations elsewhere in Golfo de Penas or north of Taitao Peninsula do not lead to carcasses in this region (Figs. 12B–12D). Although the inlets themselves are not resolved in the drift model, the net seaward surface outflow of a fjord would only allow carcasses to collect toward its head (as observed) if wind and waves in that direction dominated their drift, or if they died close to the site where they were found. Modeled winds were occasionally toward the head of Seno Newman, on Mar 20 and during Apr 14–18, but almost never in the case of Estero Slight (Fig. 16), so it is highly likely that the carcasses found within these inlets were the result of mortality within the inlets themselves. Carcasses from within these inlets could, however, be exported to nearby coastal waters and then distributed around Golfo de Penas as seen in the drift simulations for a source in Golfo Tres Montes (Fig. 12A), so mortality within the inlets of Seno Newman and Estero Slight could have been the source for carcasses found elsewhere in Golfo Tres Montes or Golfo de Penas.

Figure 16 Wind roses at the entrance to two inlets, Seno Newman (A) and Estero Slight (B), derived from a local high-resolution implementation of the WRF model.

Spoke lengths indicate the frequency of occurrence of winds from each direction. Colors represent speed. Seno Newman has a significant up-inlet component (winds from SSW) but Estero Slight does not (winds from NNE).

The accumulation of carcasses in the convoluted and extremely shallow Estero Escondido is similarly unresolvable by the drift model, but it also appears highly likely that these carcasses resulted from mortality within the inlet itself. It is, however, unclear why dozens of large whales would swim into a narrow inlet which in most parts is only between 2 and 7 m deep (maximum depth 15 m just inside extremely shallow entrance) (Fig. 17).

Figure 17 Nautical maps of Escondido and Slight Inlet.

(A) Section of the Bahia San Quintin showing Escondido Inlet (maximum depth 15 m). (B) Section of Hoppner Bay showing Estero Slight (maximum depth 152 m). Sources: Map nr 8820 and 8810 from armada de Chile. Newman Inlet is poorly charted with only five depths indicated along the inlet, the largest being 82 m.

Drift predictions from sources within Golfo de Penas, or to the south (Figs. 12A, 12C and 12D), never led to carcasses on or to the north of Taitao Peninsula, therefore the observed carcasses on the exposed shoreline in that region (Estero Cono) likely originated close to shore, either locally or to the north. The carcasses found between the Southern end of Golfo de Penas and 49°S either died very close to where they washed ashore or were transported from the large concentrations in Golfo de Penas by clockwise flow within the gulf. The five whales between 49°S and 51°S probably died locally.

Surveys in the Golfo de Penas area have sighted sei whales in all seasons, with up to 600 individuals, some even near to the shore of Golfo de Penas and Estero Slight (Table 8). Therefore, the number of whales that have been exposed to toxins could be considerable. It has been calculated that less than 10% of the gray whales that are estimated to die each year in the eastern North Pacific are washed ashore, while most sink and do not resurface (Rugh et al., 1999). Assuming a similar ratio, our observations may greatly underestimate the actual magnitude of this mortality event. Many whales may have sunk and never re-surfaced, and a significant number of carcasses may have been washed ashore on the many remote beaches that could not be surveyed due to adverse weather conditions. Others may have been destroyed by wave action from winter storms on the high-energy rocky shores that dominate the area.

In other reported MMEs, the period of the time of a massive mortality was determined by considering the number of carcasses, and their temporal and spatial extent. This ranged from two years (gray whales; Gulland et al., 2005) to a few weeks (humpback whales; Geraci et al., 1989). To determine the time span of this MME, the classification of carcasses was carried out following the disarticulation sequence proposed by Schäfer (1972).

Time since death and time of transportation at sea of the carcass are slightly different in terms of articulation and state of decomposition. Following Schäfer (1972), the first breakage of the outer tissue of a carcass at sea should occur within a week to a month, although in Chilean Patagonia the time span could be a little greater due to the low temperature. In addition, some carcasses could have drifted for some weeks, arrived intact on shore, and then decayed more rapidly exposing the bones, while other carcasses could have floated longer until skull, tail and limbs were disarticulated, but decayed more slowly due to the colder water temperatures. This was in agreement with the comparisons of the disarticulation of carcasses in the field assessed through the photographs of the different expeditions to the same area (Estero Slight, in Apr and May 2015). Nevertheless, at the present assemblage, the time until the bones were exposed was extended from one to around three months (Class 1) and time of disarticulation was shifted from three to six months (Class 2), due to the low average temperature in the study area.

Considering available information on MMEs time scales, it is reasonable to suppose this event occurred over a time span of approximately three to maximum six months (Nov 2014–Apr 2015). Nevertheless, the record of other crews (Table 2) and modeled oceanographic conditions (see “Carcass drift and potential source locations,” above) point to the beginning of the die off around February at Golfo de Penas. Thus, the Class 2 carcasses would indicate another pulse of corpses arriving at the same area in a different taphonomic condition, which could suggest: (a) longer drift time/distance transport; (b) equal arrival but different time of death; or (c) higher energy environment. The classification of “time at sea” analysis suggested that drift time was in its majority the same with a similar proportion of Class A (short drift time/distance). The analysis of the anatomic positions suggests the allochthonous nature of the deposits in all assemblages (see Pyenson et al., 2014). Only two carcasses were found in a dorsal up position, which suggests live stranding.

The average density of Golfo Tres Montes assemblages is equivalent to one third of the density calculated for Cerro Ballena, a Late Miocene (∼9 Mya) fossil red tide linked assemblage of northern Chile (3,000/km2, Pyenson et al., 2014) (Table 5). However, this difference is likely to have a sampling bias since in Golfo Tres Montes and Golfo de Penas we could only could the carcasses along the coastline, but not on the seafloor.

Conclusion

The whales died at sea, close to where they beached. About 90% of the whales died during one MME (94.7% for time since death and 87% for time at sea analysis), most probably between Feb and Apr 2015. No major mortality has occurred in the same area in 2016, but mortalities in other areas cannot be excluded (see Fig. 15 for 2016 toxin levels).

Since it is likely that all or most of the affected whales were sei whales, the documented mortality may represent a significant increase over the usual death rate of Southern Hemisphere sei whales (Reilly et al., 2008). If the frequency and magnitude of MMEs increase due to climate change this would have a significant impact on the local population and threaten the recovery of this endangered species, which in the Southern Hemisphere was reduced by whaling from about 100,000 to 24,000 individuals by 1980 (Perrin, Würsig & Thewissen, 2009).

This MME and historical data suggest that, at least during years with abundant squat lobsters, the Golfo de Penas is one of the most important feeding grounds for sei whales, hosting the largest and densest known sei whale aggregations outside the polar regions.

The MME reported herein and its probable connection to El Nino-caused red tide events throughout the Eastern Pacific could indicate that marine mammals are among the first oceanic megafauna victims of global warming.

Discoveries of dead whales in this remote area are chance finds. To clarify the extent, frequency and magnitude of MMEs, an assessment and systematic monitoring of whale populations in Central Chilean Patagonia is necessary. We suggest to do this through regular satellite images.

Supplemental Information

Supplemental Information 1 National Fisheries Service cruise report.

Click here for additional data file.

We particularly thank the organizers and participants of the expedition organized by the Chilean Fisheries Service (SERNAPESCA), especially B. Caceres, G. Garrido, M. Ulloa, F. Viddi, J. Acevedo, T. García, C. Calderón and L. Bedriñana. Thanks also to R. Brownell, N. Pyenson, L. Pastene, E. Poulin, F. Beaujot, U. Pörschmann, P. Pascoe, S. Kraft, K.-L. Pashuk and V. Beasley. Thanks to Bidema PDI, Fiscalía de Aysén and Armada de Chile for field support. We thank Percy Ramirez, Romulo Melo Cuevas, Brice Monégier, Sven Nielsen, and Regina Maria Fischer for reports of whale carcasses. We are thankful to many more people for assisting with fieldwork, technical support and logistics, for sharing or facilitating data and information, and for discussions. This is publication number 134 of Huinay Scientific Field Station.

Additional Information and Declarations

Competing Interests

Author Contributions

Animal Ethics

Field Study Permissions

Data Deposition

The authors declare that they have no competing interests.

Verena Häussermann conceived and designed the experiments, performed the experiments, analyzed the data, contributed reagents/materials/analysis tools, wrote the paper, prepared figures and/or tables and reviewed drafts of the paper, literature review, summarizing data, carried out field work in April 2015 and June 2015.

Carolina S. Gutstein conceived and designed the experiments, performed the experiments, analyzed the data, contributed reagents/materials/analysis tools, wrote the paper, prepared figures and/or tables and reviewed drafts of the paper, literature review, summarizing data, carried out field work in June 2015, did taphonomic analysis, examined ear bone.

Michael Bedington conceived and designed the experiments, performed the experiments, analyzed the data, wrote the paper, prepared figures and/or tables, reviewed drafts of the paper and drift models, construction and running of drift models.

David Cassis conceived and designed the experiments, performed the experiments, analyzed the data, contributed reagents/materials/analysis tools, wrote the paper, prepared figures and/or tables, reviewed drafts of the paper and literature review on red tides, analyzed mytilid, plankton and stomach and intestine samples for PST and AST in 2015.

Carlos Olavarria conceived and designed the experiments, analyzed the data, wrote the paper, reviewed drafts of the paper and literature review.

Andrew C. Dale conceived and designed the experiments, performed the experiments, analyzed the data, wrote the paper, prepared figures and/or tables, reviewed drafts of the paper, literature review and collection of oceanography data, support in methodology and interpretation of drift models.

Ana M. Valenzuela-Toro conceived and designed the experiments, performed the experiments, analyzed the data, wrote the paper, prepared figures and/or tables, reviewed drafts of the paper and literature review on taphonomy, carried out field work in January to March 2016, helped with taphonomic analysis.

Maria Jose Perez-Alvarez conceived and designed the experiments, analyzed the data, wrote the paper and reviewed drafts of the paper.

Hector H. Sepúlveda conceived and designed the experiments, performed the experiments, analyzed the data and collection of oceanographic data.

Kaitlin M. McConnell performed the experiments, analyzed the data, carried out field work in April 2015, January to March and April to May 2016, analyzed mytilid, plankton and stomach and intestine samples in 2016.

Fanny E. Horwitz analyzed the data and prepared figures and/or tables, carried out field work in June 2015, helped with taphonomic analysis.

Günter Försterra conceived and designed the experiments, analyzed the data, wrote the paper and reviewed drafts of the paper, writing of article.

The following information was supplied relating to ethical approvals (i.e., approving body and any reference numbers):

The samples (genetics, ear bone, stomach/intestine content and mussels) were taken during the cruise of the National Fisheries Service. The report from the cruise is supplied as a Supplemental File.

The following information was supplied relating to field study approvals (i.e., approving body and any reference numbers):

Samples of marine invertebrates were collected under permit of Subsecretaría de Pesca y Acuicultura (R.EX. 1295 del 27.04.2016). Samples of cetaceans were authorized by SERNAPESCA, Region de Aysen (Acta Numbers 2016-11-10 and 12).

The following information was supplied regarding data availability:

The research in this article did not generate any raw data.

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
