# Peer review of "Largest baleen whale mass mortality during strong El Niño event is likely related to harmful toxic algal bloom"

_PeerJ, doi:10.7717/peerj.3123_

## Round 0.1 · original submission · Major Revisions

All the referees support publication. However, two identify areas where it could be improved, and one makes particularly detailed and comprehensive suggestions. Please attend to these carefully because I believe it will improve the reading and quality of the paper significantly.

Reviewer 1 ·

Basic reporting

This is a very descriptive paper of the discovery of dead whales in SW Chile. A description of the location , orientation etc of the whales is give. Rough measurements of PST are made leading to the suggestion that saxitoxins from HABs were responsible. There appear to have been similar mass mortalities in the past in this region. It is suggested that oceanographic conditions leading to an increase HAB frequency is related to El Nino and a potential increase in mortality of whales.

The paper is well written, the observations are interesting, and it is worthy of publication in PeerJ as is.

Experimental design

An appropriate description and analysis.

Validity of the findings

Finding are quite reasonable.

·

Basic reporting

All fine, no comments

Experimental design

Methods described with sufficient detail and information to replicate.

Analysis of stomach content: was it the same mytilids in stomach content as those shown to be toxic?

A weakness in the data set that only two stomach contents were sampled. I miss comments on how representative the stomach sampling is or why it can be assumed to be representative.

Validity of the findings

Better ref on link between stronger El Nino and more HABs (of the relevant toxin producing species). If more and stronger reference cannot be found on this relationship the main conclusion that mass mortality of marine mammals is linked to climate change (via HAB) is speculative and this should be expressed more clearly. However, having such a well described case of mass mortality of whales very likely caused by toxic microalgae it highly valuable and an important publication.

Additional comments

See attached PDF with notes

·

Basic reporting

This manuscript describes a baleen whale mass mortality event (MME) off southern Chile and aims to link the event to a harmful algal bloom (HAB), and by extension to ecosystem variation associated with El Niño. Reporting the MME and understanding its cause(s) are important goals. The authors present a substantial data set that spans diverse observational methods and numercial modeling of transport. While a clear examination of this event could be an excellent contribution, the presentation in this manuscript is too underdeveloped to achieve this. While commending the authors on their responsive research program, accomplished through effective resource leveraging and strong collaborative effort, I think the authors need to invest much more effort in making this contribution complete, better organized, and understandable. I will summarize the difficulties I had in understanding this manuscript.

The basic reporting falls far short of what is needed to effectively communicate this complex and fascinating study. I will identify specific issues for each section of the manuscript, the most important being Results and Discussion.

Introduction

The introduction is sparse and contains no background information about the study region, the HAB organism that is the focus of MME causality, or ENSO variation that is postulated as influential in the HAB event. Although there is some description of the study region in the Materials and Methods section, it does not belong there because it is neither materials nor methods.

Does the brief description of tides (Lines 80-81) relate to any method or result presented in the manuscript? If not, why is it relevant?

Materials and Methods

The materials and methods section is lightly peppered with statements of results, for example lines 75-79, 86-89, 99, 112-115, 127, 150-153, 157-159.

With two sentences, the methods describe a satellite image acquired from the Pleiades-1 satellite, and it is stated that the image was analyzed. However, specifics of this analysis are not provided, and the image does not appear to be presented anywhere in the manuscript.

It is not clearly defined how time since death and time of transportation at sea were distinguished and estimated.

The explanation of class merging is unclear (lines 143-144). The text should clearly define what the numbers mean, why they are simply in parentheses separated by a space, and why category 7 is missing from the text but not the table. The merged classes are not explained in table 2, and they don’t match what is written in the text. Same issue for lines 144-146 and associated representation in table. This is one example of results presented in the methods section.

Results & Discussion

The results include 13 figures and 7 tables, yet only four pages of double-spaced text are used to present BOTH the results and discussion, and at least half of this text is discussion. I am supportive of being succinct, but this small amount of text is inadequate to present the results, provide clear interpretations of the results, and discuss these interpretations in a larger context. Voluminous results as figures and tables are stretched across a primary manuscript and supplemental material, yet most of this is barely described or completely undescribed. The splitting of content with supplemental makes the manuscript jumpy and confusing. What is the purpose of splitting off supplemental material? Is this a leftover from formatting for a different journal?

The observed distribution of carcasses is presented in Figures 1 and 4 while the modeled distribution for one of the regions, dependent upon mortality location, is presented in Figure 6. However, the presentation of results never directly compares observational and model results. My comparison of the observational and model results suggests that the northern Penas Gulf was indicated as a primary site of mortality (model distribution most consistent with observations). However, the model results are never directly described, compared to observations, or clearly interpreted. The single sentence mentioning the model results (lines 223-225) is neither adequate nor clearly interpretable. It is a vague conclusion, the basis of which the reader cannot understand from what is presented. The subsequent sentences on location of mortality are a mix of statements that range from definite to probable to unanswerable, but none of these statements are clearly linked to the presentation of results. Figure 6, containing the model results, is called out only to reference regional bathymetry in relation to whale sinking, not to actually describe model results for four different scenarios of mortality location. The caption of Figure 6 also does not define what the red star represents; the reader can reasonably guess (mortality location?) but should not have to. Also it is not clearly defined how model whale carcass “releases” were controlled. How does a single release location become many landings?

If the authors postulate that a HAB event was linked to El Niño, why is there no presentation of long-term data that define the HAB/MME period in relation to ENSO variation? ENSO indices are readily available at least, and it seems that the authors may have access to long-term environmental data in the study region.

Figure S2, a complicated representation of long-term statistical description of wind patterns in comparison to the (equinox-referenced) year of the MME, is given a single sentence of description in the methods. This is too little information to inform the reader, and it is presented in the wrong section (it is an analysis result). The key point of this result is not stated. This reader guesses that the point is that the year of the MME was anomalous with regard to forcing of up/downwelling and its seasonal progression, compared with the envelope of variation defined by data from the years 1995-2014.

Lines 215-217: A single sentence is used to summarize the results of measurements of two toxins in mussels and whale samples. This is inadequate, and the meaning is ambiguous. Were both toxins present in both types of tissue samples? Why is there no quantitative information on toxin levels? Why is the only mention of AST in the paper to say it was present, leaving the reader to wonder whether it could have been causative in the MME, besides PST?

The statement in lines 218-220 represents a very important aspect of this study, but it is not supported by the table and figure referenced. This table is also inadequately explained.

Minor comments:
LInes 42 and 43, commas needed after “whales”
Figure 1 caption: Chilean Patagonian —> Patagonia.
Line 248: missing preposition between true and the
Line 263: What is a “squat lobster year”?
Figure S4 caption: disarticulation is misspelled

Experimental design

The research presented is original and of high scientific interest. The relevant knowledge gap is presented, along with the apparent uniqueness of the exceptional baleen whale MME. Methods for characterization of mortalities are well described and referenced. Being responsive to an unexpected event, this study may allow less definitive links between predefined research questions and research efforts. This study was admirably resourceful in acquiring relevant data, however definition of research questions and thorough presentation of results in relation to specific research questions are lacking.

Validity of the findings

It is difficult to summarize validity of the findings because of the problems in basic reporting noted previously. Conclusions about relationships between ENSO variation and the HAB are vague in the current presentation because no information about ENSO variation is presented. Conclusions about the relationship between the HAB and the MME are also unclear because two different toxin classess were identified in mussels and whales, yet no quantititative toxin information was provided, and evaluation of the potential role of one toxin (AST) appears to be missing.

Additional comments

You have a very meaningful and important contribution to make with this responsive study, and I applaud the research effort. With such diverse contributions from a large group of people, all hands on deck are needed for authoring to ensure that this excellent study can be clearly understood.

---

## Round 0.2 · Minor Revisions

Please do a final proof read and note the referee's comments.

·

Basic reporting

My initial review was extensive, and the authors' revision was excellent and thorough. I have only a few minor comments on the revision:

LInes 406-407: There is some ambiguity among defining decomposition with stages, states, and classes, not consistent among text (methods, results) and table 4. Is a “decomposition stage 4” noted on line 406 defined anywhere?

Line 543: Bullet numbering was erroneously reset.

Line 815: Disarticulation is misspelled.

Experimental design

The study effectively integrates results from unexpected opportunistic observations in a remote region with multidisciplinary regional observations and model results. The topic is strongly interdisciplinary, and this contribution is substantial.

Validity of the findings

Alternative hypotheses for the MME are clearly presented, and the probable cause (PSP) is now clearly introduced for this region and supported by the available data.

Additional comments

Well done!

---

## Round 0.3 · accepted · Accept

Thank you for making the corrections and answering the journals point about the charges for this large paper. I congratulate you on an interesting and well illustrated paper.